# Disease-related mutations in PI3Kγ disrupt regulatory C-terminal dynamics and reveal a path to selective inhibitors

Manoj K Rathinaswamy[1], Zied Gaieb[2†], Kaelin D Fleming[1†], Chiara Borsari[3], Noah J Harris[1], Brandon E Moeller[1], Matthias P Wymann[3], Rommie E Amaro[2], John E Burke[1,4]*

[1]Department of Biochemistry and Microbiology, University of Victoria, Victoria, Canada; [2]Department of Chemistry and Biochemistry, University of California San Diego, San Diego, United States; [3]University of Basel, Department of Biomedicine, Basel, Switzerland; [4]Department of Biochemistry and Molecular Biology, The University of British Columbia, Vancouver, Canada

**Abstract** Class I Phosphoinositide 3-kinases (PI3Ks) are master regulators of cellular functions, with the class IB PI3K catalytic subunit (p110γ) playing key roles in immune signalling. p110γ is a key factor in inflammatory diseases and has been identified as a therapeutic target for cancers due to its immunomodulatory role. Using a combined biochemical/biophysical approach, we have revealed insight into regulation of kinase activity, specifically defining how immunodeficiency and oncogenic mutations of R1021 in the C-terminus can inactivate or activate enzyme activity. Screening of inhibitors using HDX-MS revealed that activation loop-binding inhibitors induce allosteric conformational changes that mimic those in the R1021C mutant. Structural analysis of advanced PI3K inhibitors in clinical development revealed novel binding pockets that can be exploited for further therapeutic development. Overall, this work provides unique insights into regulatory mechanisms that control PI3Kγ kinase activity and shows a framework for the design of PI3K isoform and mutant selective inhibitors.

*For correspondence:
jeburke@uvic.ca

†These authors contributed equally to this work

Competing interests: The authors declare that no competing interests exist.

## Introduction

The phosphoinositide 3-kinase (PI3K) family of enzymes are central regulators of growth, proliferation, migration, and metabolism in a plethora of cells and tissues (*Bilanges et al., 2019*; *Madsen and Vanhaesebroeck, 2020*). PI3Ks are lipid kinases that generate the lipid second messenger phosphatidylinositol-3,4,5-trisphosphate (PIP$_3$), which is utilised downstream of cell surface receptors to regulate growth, metabolism, survival, and differentiation (*Bilanges et al., 2019*). PIP$_3$, is generated by four distinct class I PI3K catalytic isoforms separated into two groups: class IA (p110α, p110β, and p110δ) and class IB (p110γ). All class I PI3Ks are constitutively associated with regulatory subunits, and the primary difference between class IA and class IB PI3Ks is their association with unique regulatory subunits. Class IA PI3Ks bind p85-like regulatory subunits encoded by *PIK3R1, PIK3R2, PIK3R3*, and class IB PI3K bind either a p101 or a p84 (also called p87$^{PIKAP}$) regulatory subunit (*Suire et al., 2005*; *Stephens et al., 1997*; *Bohnacker et al., 2009*). The four catalytic subunits of class I PI3K isoforms have distinct expression profiles: p110α and p110β are ubiquitously expressed, and p110δ and p110γ are mainly localised in immune cells (*Bilanges et al., 2019*). All PI3K isoforms have been implicated in a variety of human diseases, including cancer, immunodeficiencies, inflammation, and developmental disorders (*Goncalves et al., 2018*; *Fruman et al., 2017*; *Burke, 2018*).

The class IB p110γ isoform encoded by *PIK3CG* is a master regulator of immune cell function. It carries out almost all its physiological functions in the cell when bound to regulatory subunits, with these complexes frequently referred to as PI3Kγ (which can be either p110γ/p101 or p110γ/p84). PI3Kγ plays important roles in the regulation of myeloid (macrophages, mast cells, neutrophils) and lymphoid (T cells, B cells, and Natural Killer cells) -derived immune cells (*Camps et al., 2005*; *Patrucco et al., 2004*; *Hirsch et al., 2000*). It regulates immune cell chemotaxis (*Hirsch et al., 2000*; *Li et al., 2000*; *Sasaki et al., 2000a*), cytokine release (*Laffargue et al., 2002*; *Collmann et al., 2013*), and generation of reactive oxygen species (*Hirsch et al., 2000*), which are important processes in both the innate and adaptive immune systems. The ability of PI3Kγ to mediate multiple immune cell functions is controlled by its activation downstream of numerous cell surface receptors, including G-protein coupled receptors (GPCRs) (*Stoyanov et al., 1995*), the IgE/Antigen receptor (*Laffargue et al., 2002*), receptor tyrosine kinases (RTKs) (*Schmid et al., 2011*), and the Toll-like receptors (TLRs) (*Luo et al., 2018*; *Luo et al., 2014*). Activation of PI3Kγ downstream of these stimuli are potentiated by their p84 and p101 regulatory subunits (*Bohnacker et al., 2009*; *Luo et al., 2018*; *Stephens et al., 1997*; *Vadas et al., 2013*; *Kurig et al., 2009*). In mouse models, loss of PI3Kγ either genetically or pharmacologically is protective in multiple inflammatory diseases including cardiovascular disease (*Patrucco et al., 2004*), arthritis (*Camps et al., 2005*), Lupus (*Barber et al., 2005*), asthma (*Collmann et al., 2013*), pulmonary inflammation and fibrosis (*Thomas et al., 2009*; *Campa et al., 2018*), and metabolic syndrome (*Breasson et al., 2017*). PI3Kγ is also a driver of pancreatic ductal adenocarcinoma progression through immunomodulatory effects (*Kaneda et al., 2016a*), and targeting PI3Kγ in the immune system in combination with checkpoint inhibitors has shown promise in experimental cancer therapy (*De Henau et al., 2016*; *Kaneda et al., 2016b*).

Extensive biophysical and biochemical assays have identified many of the molecular mechanisms underlying PI3Kγ regulation. The p110γ enzyme is composed of five domains, a putative uncharacterised adaptor binding domain (ABD), a Ras binding domain (RBD), a C2 domain, a helical domain, and a bi-lobal lipid kinase domain (*Walker et al., 1999*; *Figure 1A*). PI3Kγ activation is primarily mediated by Gβγ subunits downstream of GPCR signalling, through a direct interaction of Gβγ with the C2-helical linker of p110γ (*Vadas et al., 2013*). Activation of PI3Kγ by Gβγ requires a secondary interaction between Gβγ and regulatory subunits for physiologically relevant activation (*Stephens et al., 1997*), with the free p110γ subunit having no detectable activation downstream of GPCR activation in cells (*Deladeriere et al., 2015*). In addition, PI3Kγ activation can be facilitated by Ras GTPases interacting with the RBD of p110γ (*Pacold et al., 2000*), with the same interface also putatively mediating activation by Rab8 (*Luo et al., 2014*). Experiments exploring a novel type II-like kinase inhibitor that targets an active conformation of PI3Kγ revealed novel molecular aspects of regulation involving the C-terminal regulatory motif of the kinase domain, which is composed of the kα7, 8, 9, 10, 11, 12 helices that surround the activation loop, and keep the enzyme in an inhibited state (*Gangadhara et al., 2019*; *Figure 1B*). The kα10, kα11, and kα12 helices are sometimes referred to as the regulatory arch (*Vadas et al., 2011*). Inhibition mediated by the C-terminal regulatory motif is conserved through all class I PI3Ks, although for all other isoforms, this inhibited conformation requires interactions with a p85 regulatory subunit (*Figure 1—figure supplement 1A +B*; *Burke, 2018*). The activation of all class I PI3Ks is proposed to require a conformational change in the regulatory motif leading to a reorientation of the C-terminus to a conformation that is compatible with membrane binding (*Figure 1—figure supplement 1C*; *Burke, 2018*). The p110γ catalytic subunit is unique in that it maintains an inactive conformation in the absence of regulatory subunits. This conformation is proposed to be maintained by a Tryptophan lock, where W1080 maintains a closed conformation of the membrane binding C-terminal kα12 helix, leading to an inactive conformation of the activation loop (*Gangadhara et al., 2019*; *Figure 1B+D*).

Disruption of PI3K signalling by either activating or inactivating mutations and deletions are involved in multiple human diseases. Overexpression of any activated class I PI3K isoform can lead to oncogenic transformation (*Kang et al., 2006*), although p110α is the most frequently mutated in human disease. Activating p110α mutations are linked to both cancer (*Samuels et al., 2004*; *Vasan et al., 2019*) and overgrowth disorders (*Lindhurst et al., 2012*), while activating p110δ mutations are linked to primary immuno-deficiencies (*Dornan et al., 2017*; *Lucas et al., 2016*; *Angulo et al., 2013*). A high proportion of these activating mutations cluster to the C-terminal regulatory motif of the p110 catalytic subunits. Multiple p110γ mutations have been identified in cancer

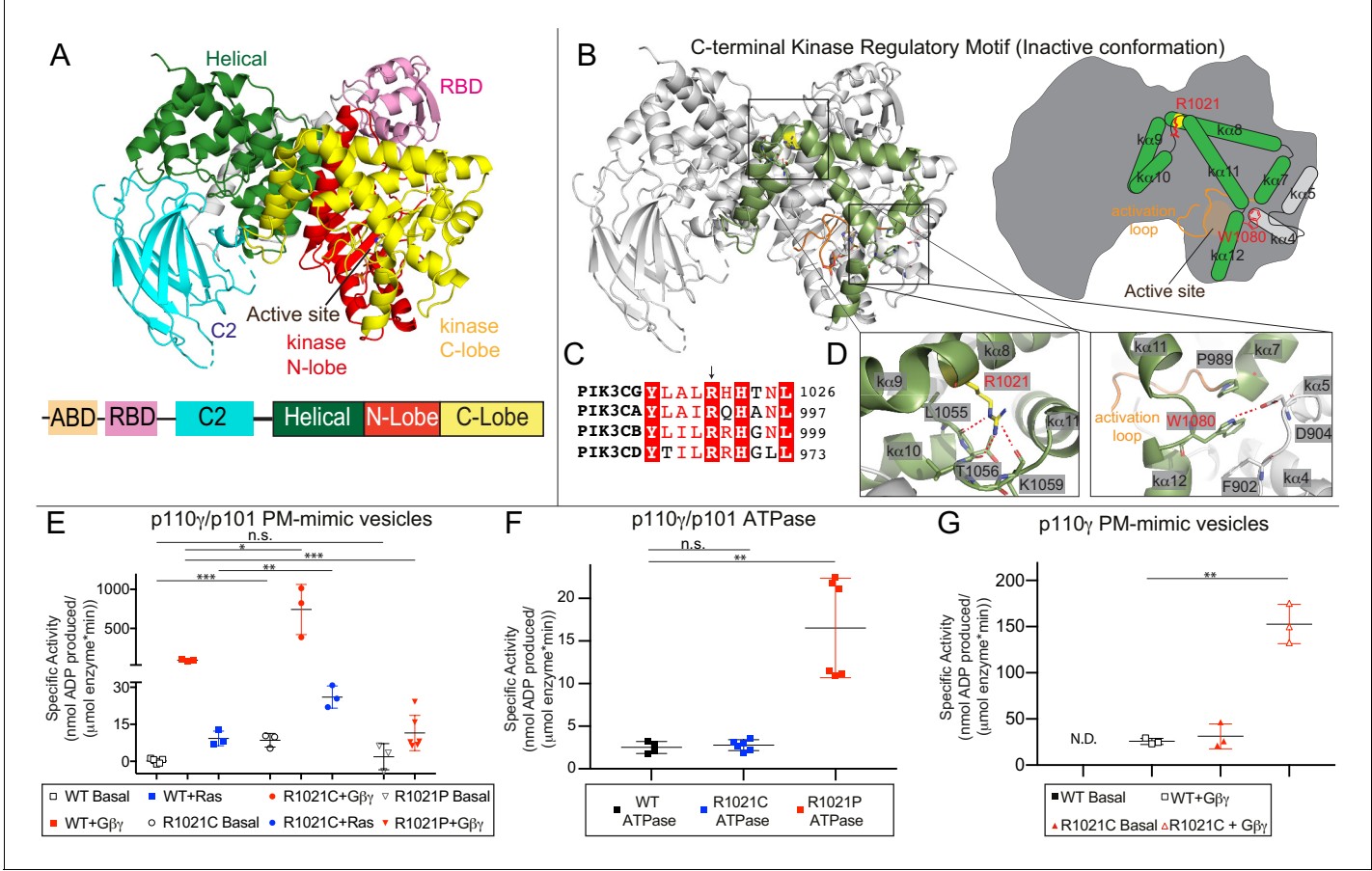

**Figure 1.** Mutations in the regulatory C-terminal motif of the kinase domain alter PI3Kγ activity. (**A**) Domain architecture of p110γ (PDB ID: 6AUD) (*Safina et al., 2017*), with the domain schematic shown beneath. (**B**) Model of the C-terminal regulatory motif of the kinase domain of p110γ. The helices that make up the regulatory motif, which includes the regulatory arch (kα10, 11,12) and those that pack against them (kα7, 8, 9) are highlighted in green both in the structural model and cartoon schematic. (**C**) Alignment of residues around R1021 in p110γ with class IA p110 isoforms. (**D**) A close up of the W1080 'Tryptophan lock' interaction with kα7 and the kα4-kα5 loop which maintains an inhibited conformation is shown, as well as the interaction of the R1021 residue with residues on the kα10-kα11 loop. (**E**) Lipid kinase activity assays testing the activity of WT, R1021C, and R1021P p110γ/p101 WT basally and in the presence of lipidated Gβγ and HRas. Experiments were carried out with 50–3000 nM kinase, 1500 nM Ras, 1500 nM Gβγ, all in the presence of 100 μM ATP and 1 mg/mL PM-mimic vesicles 5% phosphatidylinositol 4,5 bisphosphate (PIP₂), 20% phosphatidylserine (PS), 10% phosphatidyl choline (PC), 50% phosphatidylethanolamine (PE), 10% Cholesterol, 5% sphingomyelin (SM). (**F**) Activity assays testing the intrinsic ATPase activity (ATP conversion in the absence of membrane substrate) for wild type and mutant p110γ/p101 complexes. (**G**) Lipid kinase activity assays testing the activity of WT and R1021C for the free p110γ catalytic subunit with and without lipidated Gβγ. Lipid kinase activity was generated by subtracting away non-specific ATPase activity, for unstimulated WT p110γ there was no detectable lipid kinase activity above basal ATPase activity (N. D.). For panels E-G every replicate is plotted, with error shown as standard deviation (n = 3–6). Two tailed p-values represented by the symbols as follows: ***<0.001; **<0.01; *<0.05; n.s. > 0.05. Refer to the *Figure 1—source data 1* file for the specific activity values.

The online version of this article includes the following source data and figure supplement(s) for figure 1:

**Source data 1.** Summary of HDX-MS data sets (see attached excel source data files).

**Figure supplement 1.** Comparing the different regulatory mechanisms that maintain the C-terminal regulatory motif in a inhibited state in the class I PI3Ks.

**Figure supplement 2.** Purification of mutated p110γ / p101 complexes.

patients (*Lowery et al., 2019*; *AACR Project GENIE Consortium, 2017*; *Tate et al., 2019*), although at a lower frequency than p110α mutations. It would be expected that these mutations are activating, although this has not been fully explored. Intriguingly, p110γ loss-of-function mutations in the C-terminal regulatory motif (R1021P, N1085S) have been identified in patients with immunodeficiencies (*Takeda et al., 2019*; *Thian et al., 2020*; *Figure 1B*). PI3K mediated diseases being caused by both activating and inactivating mutations highlights the critical role of maintaining appropriate PIP₃ levels for human health.

The involvement of activated PI3K signalling in multiple diseases has motivated class I PI3K inhibitor development. There are, however, toxicity effects associated with compounds that target all PI3K isoforms by mechanism-based adverse side effects (*Fruman and Rommel, 2014*), driving the development of isoform selective inhibitors. These efforts have led to multiple clinically approved inhibitors of PI3Kα and PI3Kδ (*André et al., 2019*; *Brown et al., 2014*; *Flinn et al., 2014*). The critical role of PI3Kγ in inflammation and the tumour microenvironment has stimulated development of PI3Kγ specific inhibitors. Two main strategies for generating p110γ selective ATP-competitive inhibitors have been established: (i) targeting p110γ-specific pockets in and around the ATP-binding site which are not conserved among p110 isoforms (*Collier et al., 2015*; *Evans et al., 2016*), and (ii) targeting selective p110γ conformational changes (*Gangadhara et al., 2019*). Intriguingly, the conformational selective p110γ inhibitors appear to target its putatively activated conformation.

The parallel discovery of disease linked mutations in the C-terminal regulatory motif, and conformational selective p110γ inhibitors that cause altered dynamics of the C-terminus led us to investigate the underlying molecular mechanisms. Using a combined biochemical and biophysical approach, we characterised the dynamic conformational changes caused by the loss-of-function R1021P p110γ mutation, as well as a putative oncogenic R1021C p110γ mutation identified in the Catalogue of Somatic Mutations in Cancer database (COSMIC; *Tate et al., 2019*). We found that the R1021P mutant leads to greatly decreased protein stability, while the activating R1021C mutation leads to localised conformational disruption of the regulatory motif at the c-terminus. A screen of a number of PI3Kγ selective and pan-PI3K inhibitors revealed that many of these molecules induced allosteric conformational changes in p110γ. A combined X-ray crystallography and hydrogen deuterium exchange mass spectrometry (HDX-MS) approach showed that inhibitor interactions with the activation loop mediates the allosteric conformational changes. Intriguingly, similar conformational changes occurred with the R1021C mutant. The R1021C mutant was also found to be more sensitive to inhibition by kinase inhibitors that cause allosteric conformational changes. Overall, this work provides a unique insight into how mutations alter PI3Kγ regulation, and paves the way to novel strategies for isoform and mutant selective PI3K inhibitors.

## Results

### R1021P and R1021C mutations alter the activity of p110γ

The recent discovery of an inactivating disease-linked mutation in *PIK3CG* located near the C-terminus of the p110γ kinase domain (R1021P) in immunocompromised patients led us to investigate the molecular mechanism of this mutation. Intriguingly, this same residue is found to be mutated in the COSMIC database (R1021C) (*Tate et al., 2019*). The R1021 residue in p110γ is conserved across all class I PI3K isoforms (*Figure 1C*), and mutations in the equivalent residue in *PIK3CA* (p110α) are putatively oncogenic (R992L/N) (*Tate et al., 2019*). To define the effect of these mutations on protein conformation and biochemical activity, we generated them recombinantly in complex with the p101 regulatory subunit. Both the p110γ R1021C and R1021P complexes with p101 eluted from gel filtration similar to wild-type p110γ-p101, suggesting they were properly folded (*Figure 1—figure supplement 2*). However, the yield of the R1021P complex with p101 was dramatically decreased relative to both wild-type and R1021C p110γ, indicating that this mutation may decrease protein stability, consistent with decreased p110γ and p101 expression in patient tissues (*Takeda et al., 2019*). We also generated the free R1021C p110γ subunit, however we could not express free p110γ R1021P, further highlighting that this mutation likely leads to decreased protein stability.

The R1021 residue forms hydrogen bonds with the carbonyl oxygens of L1055, T1056, and K1059 located in or adjacent to the regulatory arch helices kα10 and kα11 of PI3Kγ (*Figure 1D*). Both R1021C and R1021P would be expected to disrupt these interactions, with the R1021P also expected to distort the secondary structure of the kα8 helix. The R1021P has been previously found to lead to greatly decreased lipid kinase activity in vitro (*Takeda et al., 2019*). To characterise these mutations, we carried out biochemical assays of wild-type, R1021C, and R1021P p110γ/p101 complexes against plasma membrane-mimic lipid vesicles containing 5% PI$P_2$. Assays were carried out in the presence and absence of lipidated Gβγ and Ras. Lipidated Gβγ leads to a > 100-fold activation of lipid kinase activity for wild-type complexes, with Ras causing a ~ 10-fold activation. These assays revealed that p110γ/p101 R1021C was ~eightfold more active than wild-type both basally and in the

presence of Gβγ, and ~two- to threefold more active upon Ras activation (*Figure 1E*). The R1021C mutant also showed a ~eightfold increase in lipid kinase activity compared to wild-type when assaying the free 110γ subunit (*Figure 1G*). This is consistent with the R1021C mutation destabilising the regulatory motif, potentially disrupting the Tryptophan lock and leading to increased membrane recruitment.

The R1021P complex showed weak but detectable basal lipid kinase activity; however, there was almost no activation by lipidated Gβγ compared to wild-type, suggesting this complex would be almost completely inactive in a physiological context (*Figure 1E*). To examine if the catalytic machinery of the R1021P mutant was intact, we carried out experiments looking at ATPase activity (nonproductive turnover of ATP in the absence of lipid substrate). Intriguingly, R1021P p110γ-p101 showed higher basal ATPase activity compared to wild-type p110γ/p101 and R1021C p110γ-p101, revealing that it still has catalytic activity, but almost no activity on membrane substrate (*Figure 1F*). This suggests that the minimal structural features required for ATP hydrolysis are maintained in the R1021P mutant, but that either membrane binding or PI$P_2$ recognition is severely impaired.

## R1021P and R1021C cause allosteric conformational changes throughout the regulatory C-terminal motif

We carried out hydrogen deuterium exchange mass spectrometry (HDX-MS) experiments to define the molecular basis for why two different mutations at R1021 have opposing effects on lipid kinase activity. HDX-MS is a technique that measures the exchange rate of amide hydrogens, and as the rate is dependent on the presence and stability of secondary structure, it is an excellent probe of protein conformational dynamics (*Vadas et al., 2017*). HDX-MS experiments were performed on complexes of wild-type p110γ/p101, R1021C p110γ/p101, and R1021P p110γ/p101, as well as the free wild-type and R1021C p110γ. The coverage map of the p110γ and p101 proteins was composed of 153 peptides spanning ~93% percent of the exchangeable amides (*Supplementary file 1*).

The R1021C and R1021P mutations led to significant changes in the conformational dynamics of the p110γ catalytic and p101 regulatory subunits (*Figure 2A–G*). The R1021P mutation resulted in large increases in exchange throughout almost the entire C2, helical, and kinase domains (*Figure 2B +D*). Comparing the rates of hydrogen exchange between wild-type, R1021C, and R1021P showed many regions where both R1021C and R1021P mutations caused increased H/D exchange. However, for the majority of these regions, the R1021P mutation led to increased exchange at early (3 s) and late timepoints (3000 s) of exchange, indicating that this mutation was leading to significant disruption of protein secondary structure (*Figure 2G*). This large-scale destabilisation throughout the protein may explain the low yield and decreased kinase activity.

The R1021C mutation resulted in increased H/D exchange in the C2, helical and kinase domains of p110γ. Intriguingly, many of the changes in dynamics of the helical and kinase domains are similar to those observed upon membrane binding (*Vadas et al., 2013*). The largest differences occurred in the helices in the C-terminal regulatory motif (kα7–12) (*Figure 2A+C*). A peptide spanning the C-terminal end of the activation loop and kα7 (976–992) showed increased exchange, with these changes primarily occurring at later timepoints of exchange (3000 s) (*Figure 2G*). This is indicative of these regions maintaining secondary structure, although with increased flexibility. These increases in exchange for the R1021C mutant were conserved for the free p110γ subunit, although with larger increases in exchange compared to the p110γ/p101 complex (*Figure 2—figure supplement 1A–C*). Intriguingly, we found larger decreases in exchange with the p101 subunit for the R1021C p110γ compared to WT p110γ, which suggests a global stabilisation by the p101 subunit that mitigates some of the effects of the R1021C mutant (*Figure 2—figure supplement 1E+F*). Differences in HDX-MS with R1021C is consistent with a disruption of the inhibitory conformation of the regulatory motif. In addition, the increased dynamics of the activation loop seen in the mutant may pre-organise the catalytic machinery for activation. This is consistent with previous HDX-MS analysis of the regulatory mechanisms of class IA PI3Ks which revealed that increased dynamics of the activation loop occurred concurrently with increased lipid kinase activity (*Dornan et al., 2017*; *Dornan and Burke, 2018*; *Burke and Williams, 2013*; *Burke et al., 2012*; *Burke et al., 2011*).

The two mutations in R1021C and R1021P both caused increased exchange in the p101 subunit. Peptides spanning 602–623, and 865–877 of p101 showed similar increases in exchange for both R1021C and R1021P, with R1021P also leading to increased exchange in a peptide nearer the N-terminus (102-122) (*Figure 2E+F*, S3D). As there is no structural model for the p101 subunit, it is hard

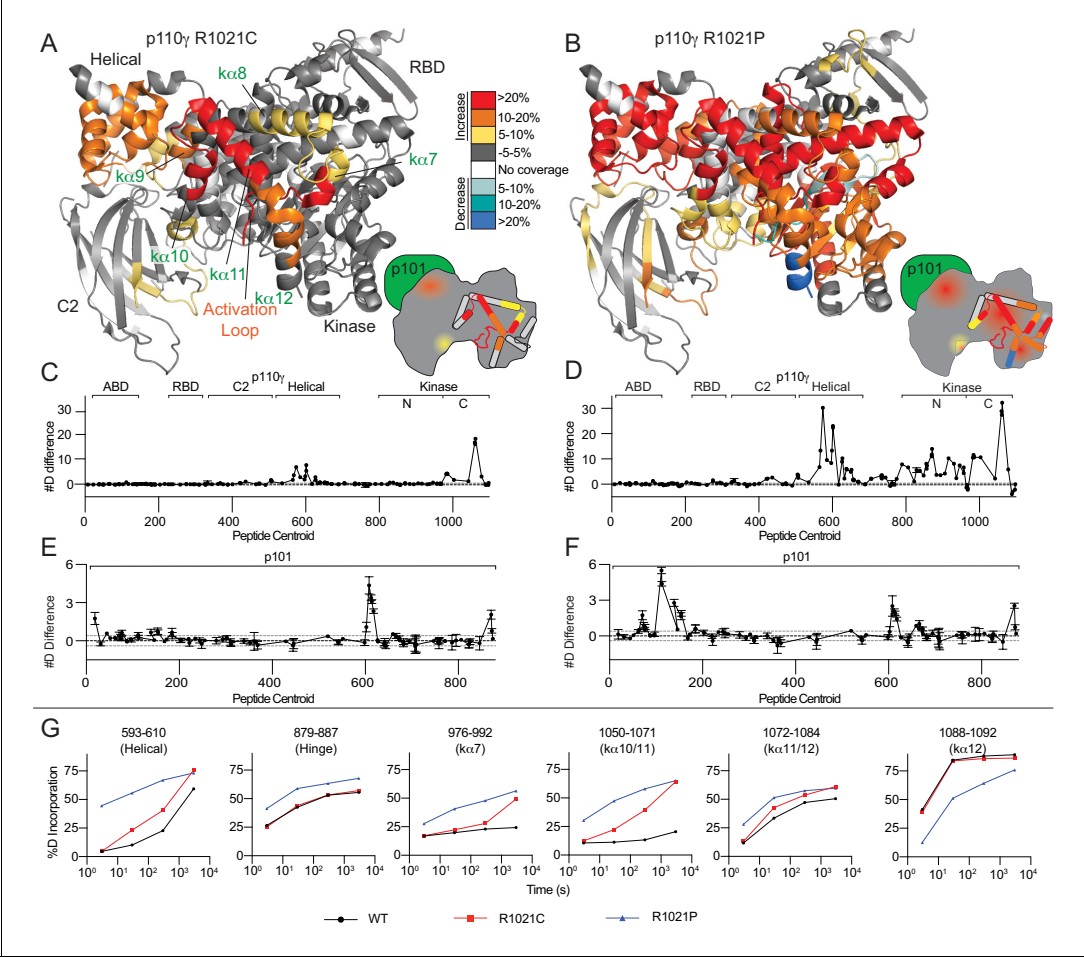

**Figure 2.** R1021C and R1021P mutations in p110γ are destabilising, with R1021P leading to global destabilisation and R1021C leading to localised disruption of the C-terminal regulatory W1080 Tryptophan 'lock'. (A+B) Peptides showing significant deuterium exchange differences (>5%,>0.4 kDa and p<0.01 in an unpaired two-tailed t-test) between wild-type and R1021C (A) and wild-type and R1021P (B) p110γ/p101 complexes are coloured on a model of p110γ (PDB: 6AUD)(*Safina et al., 2017*). Differences in exchange are coloured according to the legend. (C+D) The number of deuteron difference for the R1021C and R1021P mutants for all peptides analysed over the entire deuterium exchange time course for p110γ. Every point represents the central residue of an individual peptide. The domain location is noted above the primary sequence. A cartoon model of the p110γ structural model is shown according to the legend in panels A+B. Error is shown as standard deviation (n = 3). (E+F) The number of deuteron difference for the R1021C and R1021P mutants for all peptides analysed over the entire deuterium exchange time course for p101. Every point represents the central residue of an individual peptide. Error is shown as standard deviation (n = 3). (G) Selected p110γ peptides that showed decreases and increases in exchange are shown. The HDExaminer output data and the full list of all peptides and their deuterium incorporation is shown in the *Figure 2—source data 1* file.

The online version of this article includes the following source data and figure supplement(s) for figure 2:

**Source data 1.** Source data for HDX-MS comparing p110g/p101 and mutants.

**Figure supplement 1.** Differences in HDX for the R1021C mutation in free p110γ.

**Figure supplement 1—source data 1.** Source data for HDX-MS comparing free p110g and free p110g R1021C.

to unambiguously interpret this data; however, these may represent increased H/D exchange due to partial destabilisation of the complex.

## Molecular dynamics of p110γ R1021C and R1021P mutants

We carried out Gaussian-accelerated Molecular Dynamics (GaMD) simulations of wild-type p110γ and its R1021C and R1021P variants to provide additional insight into the underlying molecular mechanisms of altered H/D exchange behaviour and different lipid kinase activity. Using the crystallographic structure of p110γ lacking the N-terminus [amino acids 144–1102, PDB: 6AUD

(*Safina et al., 2017*)], we generated the activation loop and other neighbouring loops as described in the methods, removed the co-crystallised ligand, and mutated R1021 to a cysteine and proline, resulting in three systems: WT, R1021C, and R1021P. Three replicas of fully solvated all-atom GaMD simulations were run for each model with AMBER18 achieving a cumulative extensive sampling of ~3, ~4.1, and ~1.5 µs for WT, R1021C, and R1021P, respectively (*Figure 3A+B*).

To quantify the effect of mutations on the structural dynamics of p110γ, we calculated the root-mean-square-fluctuation (RMSF) of residues neighbouring the mutation site. RMSF was calculated to determine average flexibility of each residue's Cα and Cβ atoms around their mean position (*Figure 3C*). This revealed increased fluctuations in the residues forming the loop between kα10 and kα11 in the mutated systems, specifically residues T1056, V1057, and G1058 at the C-terminus of kα10. Many of these residues participate in hydrogen bonds with R1021 in WT (*Figure 3B*).

Analysis of the simulations revealed that mutation of R1021 results in disruption of the hydrogen bonding network between R1021 and L1055, T1056, and K1059 in the kα10-kα11 region. There were also alterations in the intra and inter helix hydrogen bonds in kα8, kα9, kα10, and kα11 (*Figure 3D*, S4). Hydrogen bonding occupancies between Y1017 and T1056 decreased from 71% in WT to 56% and 45% in the R1021C and R1021P systems, respectively. Examining the kα8-kα9 backbone hydrogen bonding at the site of mutation, both mutations showed a disruption between C/P1021 and T1024. Additionally, the proline mutation showed complete disruption of backbone hydrogen bonds at A1016-L1020 and Y1017-P1021, decreased bonding occupancy at K1015-A1019

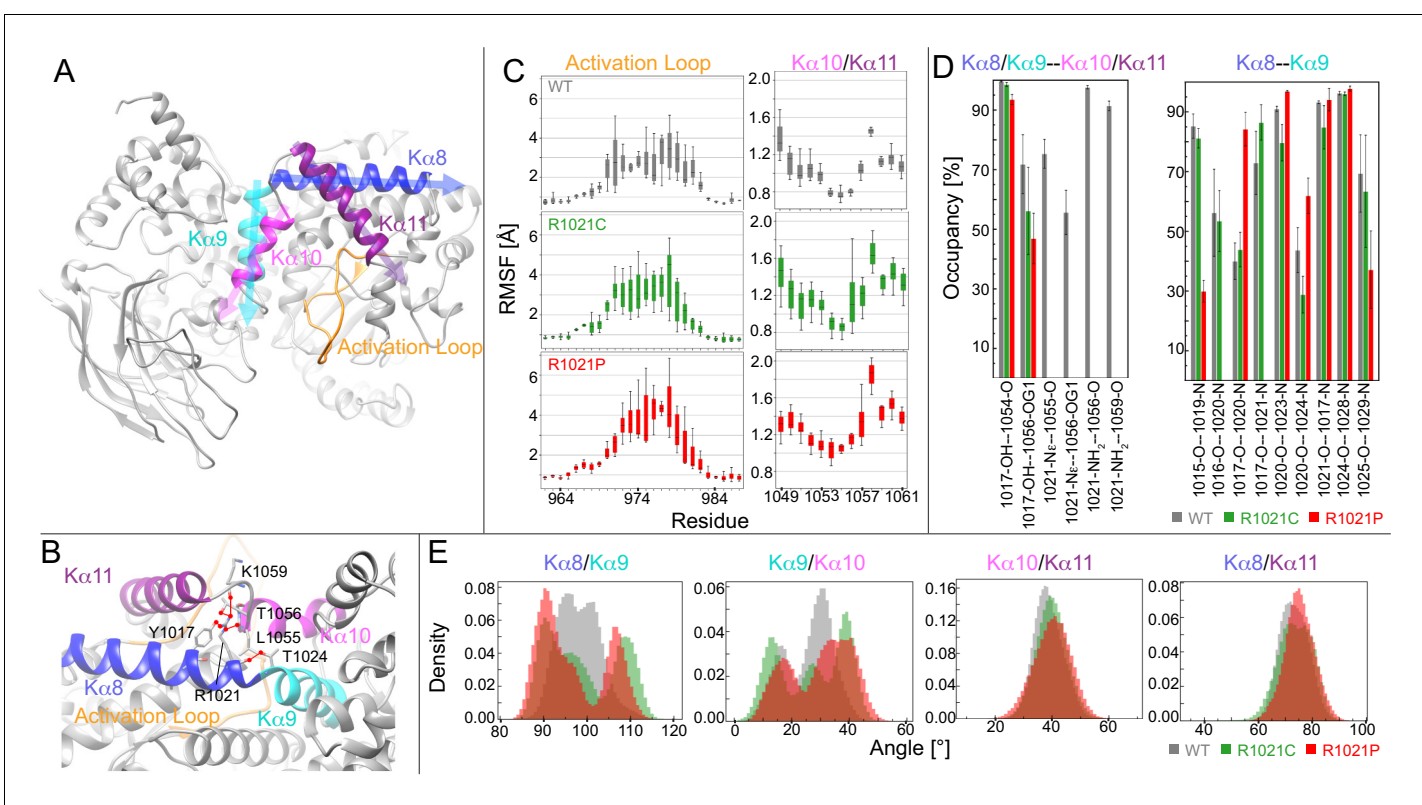

**Figure 3.** Molecular dynamic simulations reveal that the R1021C and R1021P mutations show increased instability in p110γ. (**A**) Model of p110γ showcasing the regulatory domain's kα8 (995–1023), kα9 (1024–1037), kα10 (1045–1054), and kα11 (1057–1078) helices, and the activation loop (962-988). (**B**) A zoomed-in snapshot of R1021 microenvironment showing residues in licorice. Hydrogen bonds with R1021 are drawn as red lines. (**C**) RMSF [Å] of each residue's Cα and Cβ atoms in the activation loop and the kα10/kα11 helices, respectively. RMSF values for each atom across replicates are shown as a quantile plot, with error shown as standard deviation (n = 3). (**D**) The mean and standard deviation of hydrogen bond occupancies between the indicated helices/sets of helices across replicates (n = 3). (**E**) Inter-angle density distributions across all replicas between kα8, kα9, kα10, and kα11. In all panels, WT, R1021C, and R1021P are coloured in grey, green, and red, respectively.

The online version of this article includes the following figure supplement(s) for figure 3:

**Figure supplement 1.** Differences between molecular dynamic simulations of WT, R1021C, and R1021P.

and N1025-I1029, and increased bonding occupancy of Y1017-L1020 and P1021-T1024. The notable increase in hydrogen bonding disruption in the R1021P compared to R1021C could be responsible for the increased destabilisation observed by HDX-MS.

To obtain further insights into the dynamic behaviour of the C-terminus of the kinase domain and how mutation of R1021 alters conformational dynamics, we monitored the fluctuations of four different angles formed between kα8, kα9, kα10, and kα11 (*Figure 3E*). The simulations revealed increased angle fluctuations in the mutant simulations between kα8 and kα9, and kα9 and kα10, with a bimodal distribution in the kα8/kα9 angle compared to WT. There was also increased fluctuations in the activation loop in the mutants compared to WT (*Figure 3C*, *Figure 3—figure supplement 1*). We did not observe large-scale conformational disruption in the R1021P mutant, which may require more extensive MD timescales to observe global stability differences. Overall, the MD, HDX-MS, and kinase assays are consistent with the R1021C mutation altering the dynamics of the regulatory motif, and destabilizing the region around the tryptophan lock. This region must open to allow for membrane binding (*Vadas et al., 2013*), suggesting that this mutation may pre-organise the regulatory motif for membrane binding/catalysis.

## Multiple PI3Kγ inhibitors lead to allosteric conformational changes

The increased H/D exchange in the regulatory motif for the R1021C mutant is similar to previously observed differences caused by cyclopropyl ethyl containing isoindolinone inhibitors (*Gangadhara et al., 2019*). This result suggested it would be worth interrogating the dynamics of p110γ induced by multiple classes of p110γ inhibitors. Specifically, we wanted to understand if inhibitors that promote allosteric conformational changes might be useful in the development of isoform/mutant selective compounds.

We performed HDX-MS experiments with seven potent PI3K inhibitors on free p110γ to define the role of allostery in PI3Kγ inhibition. We analysed inhibitors that were selective for PI3Kγ (AS-604850 [*Camps et al., 2005*], AZ2 [*Gangadhara et al., 2019*], NVS-PI3-4 [*Collmann et al., 2013*; *Bruce et al., 2012*], and IPI-549 *Evans et al., 2016*) as well as pan-PI3K inhibitors (PIK90 [*Knight et al., 2006*], Omipalisib [*Knight et al., 2010*], and Gedatolisib *Venkatesan et al., 2010*). Of these compounds, only AS-604850, PIK90, and Omipalisib have been structurally characterised bound to p110γ. A table summarizing these compounds and their selectivity for different PI3K isoforms is shown in *Supplementary file 5*. Deuterium exchange experiments were carried out with monomeric p110γ over 4 timepoints of deuterium exchange (3,30,300, and 3000 s). We obtained 180 peptides covering ~89% percent of the exchangeable amides (*Supplementary file 2*). To verify that results on the free p110γ complex are relevant to the physiological p110γ/p101 complex, we also carried out experiments with the p110γ/p101 complex with Gedatolisib and IPI-549, with the free p110γ showing almost exactly the same differences as seen for the p110γ/p101 complex (*Figure 4—figure supplement 1*).

Based on the H/D exchange differences observed with inhibitors present, we were able to classify the inhibitors into three broad groups. The first group contains the isoquinolinone compound IPI-549, the imidazo[1,2 *c*]quinazoline molecule PIK-90 and the thiazolidinedione compound AS-604850 (*Figure 4A+B*). These compounds caused decreased exchange near the active site, with the primary protected region being the hinge region between the N- and C- lobes of the kinase domain (*Figure 4B+E*). No (IPI-549, AS-604850) or very small (PIK-90) increases in deuterium incorporation were observed (*Figure 4A*, *Figure 4—figure supplement 2A–C*), suggesting that there are localised allosteric conformational changes for these compounds.

The H/D exchange experiments revealed a second class of inhibitors that showed decreased exchange at the active site, but also significant increases in exchange in the kinase and helical domains (*Figure 4A+C*, *Figure 4—figure supplement 2D–F*). The second group includes the bis-morpholinotriazine molecule Gedatolisib, difluoro-benzene sulfonamide compound Omipalisib, and the PI3Kγ-specific thiazole derivative NVS-PI3-4. Binding of these inhibitors caused increased exchange in multiple regions of the kinase regulatory motif, including kα7, kα10, kα11, and kα12. The peptide covering kα7 also spans the C-terminal end of the activation loop (*Figure 4C+E*). In addition to these changes, there were also increases in exchange within the helical domain. Intriguingly, for Gedatolisib, the regions of the protein that showed differences in H/D exchange matched very closely to those observed in the R1021C mutant. This suggests that the conformational changes

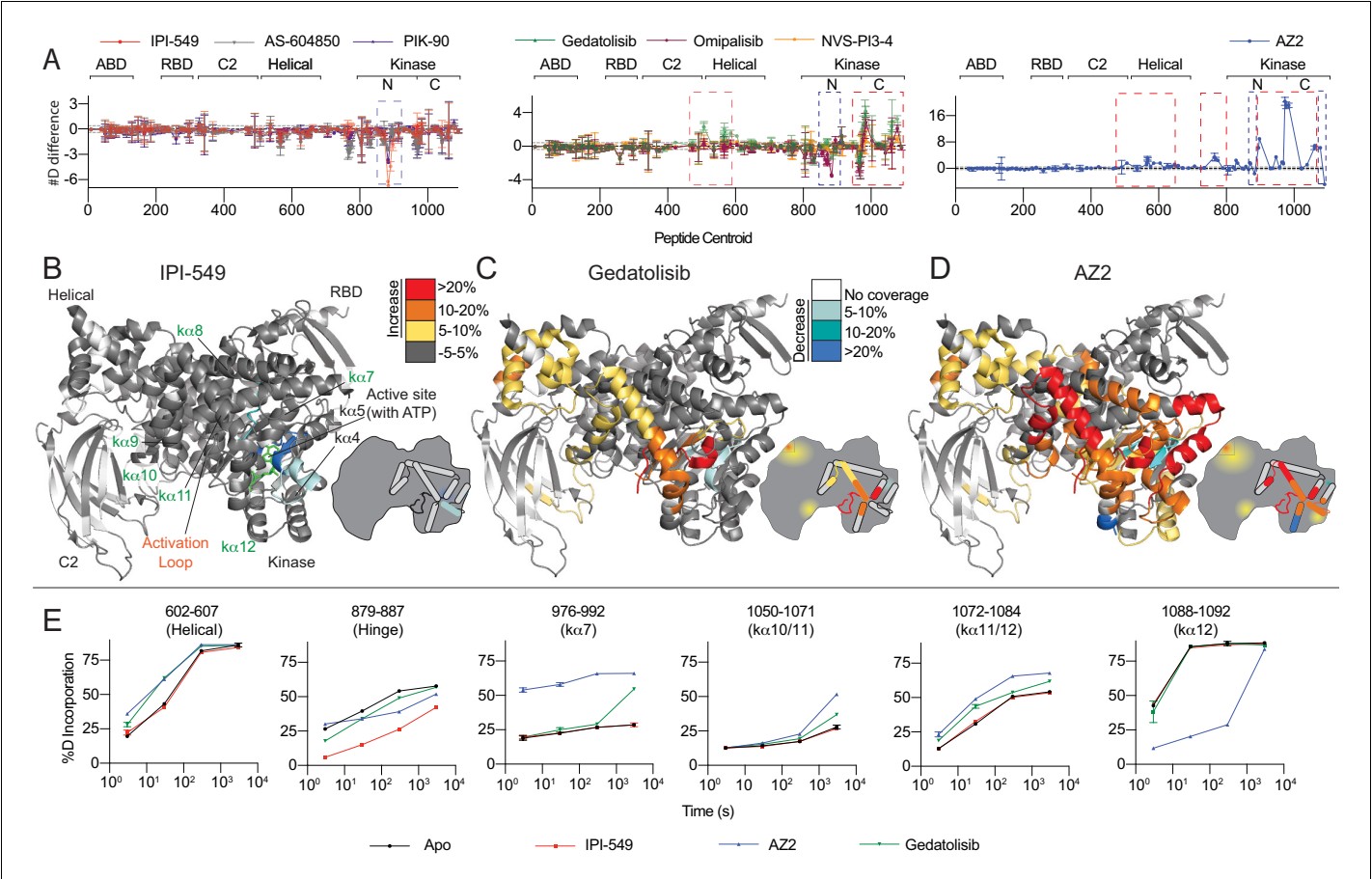

**Figure 4.** HDX-MS reveals that different classes of PI3K inhibitors lead to unique allosteric conformational changes. (A) The number of deuteron difference for the seven different inhibitors analysed over the entire deuterium exchange time course for p110γ. Every point represents the central residue of an individual peptide. The domain location is noted above the primary sequence. Error is shown as standard deviation (n = 3). (B–D) Peptides showing significant deuterium exchange differences (>5%,>0.4 kDa and p<0.01 in an unpaired two-tailed t-test) between wild-type and IPI-549 (B), Gedatolisib (C), and AZ2 (D) are coloured on a model of p110γ (PDB: 6AUD). Differences in exchange are mapped according to the legend. A cartoon model in the same format as *Figure 1* is shown as a reference. (E) Selected p110γ peptides that showed decreases and increases in exchange are shown. The HDExaminer output data and the full list of all peptides and their deuterium incorporation is shown in the *Figure 4—source data 1* file. The online version of this article includes the following source data and figure supplement(s) for figure 4:

**Source data 1.** Source data for HDX-MS comparing free p110g with and without inhibitors.
**Figure supplement 1.** Differences in HDX for free p110γ and p110γ/p101 with selected inhibitors.
**Figure supplement 1—source data 1.** Source data for HDX-MS comparing p110g/p101 with and without IPI-549/Gedatolisib.
**Figure supplement 2.** HDX-MS reveals that different classes of PI3K inhibitors lead to unique allosteric conformational changes.

induced by these compounds may partially mimic the activated state that occurs in the R1021C mutant.

Finally, AZ2 caused large-scale increased exposure throughout large regions of the helical and kinase domains (*Figure 4A+D*), consistent with previous reports (*Gangadhara et al., 2019*). The same regulatory motif regions that showed increased exchange with Gedatolisib showed much larger changes with AZ2. Importantly, increased exchange was observed at earlier timepoints for AZ2 compared to Gedatolisib (example peptide 976–992 covering the activation loop and kα7), suggesting that AZ2 leads to a complete disruption of secondary structure, with Gedatolisib likely causing increased secondary structure dynamics (*Figure 4E*).

This shows that multiple PI3K inhibitors can cause large scale allosteric conformational changes upon inhibitor binding, however, deciphering the molecular mechanism of these changes were hindered by lack of high-resolution structural information for many of these compounds.

## Structures of PI3Kγ bound to IPI-549, Gedatolisib, and NVS-PI3-4

To further define the molecular basis for how different inhibitors lead to allosteric conformational changes, we solved the crystal structure of p110γ bound to IPI-549, Gedatolisib, and NVS-PI3-4 at resolutions of 2.55 Å, 2.65 Å, and 3.15 Å, respectively (*Figure 5A–C*, *Figure 5—figure supplement 1A–F*). The inhibitor binding mode for all were unambiguous (*Figure 5—figure supplement 2A*).

These structures revealed insights into how IPI-549 and NVS-PI3-4 can achieve selectivity for PI3Kγ (*Figure 5—figure supplement 1G–H*). All inhibitors formed the critical hydrogen bond with the amide hydrogen of V882 in the hinge region, which is a conserved feature of ATP competitive PI3K kinase inhibitors. NVS-PI3-4 leads to opening of a p110γ unique pocket mediated by a conformational change in K883 (*Figure 5—figure supplement 1G+H*). The opening of K883 is accommodated by it rotating into contact with D884 and T955. This opening would not be possible in p110α and p110δ as the corresponding K883 residue (L829 in p110δ and R852 in p110α) would clash with the corresponding T955 residue (R902 in p110δ and K924 in p110α) (*Figure 5—figure supplement 1I+J*). IPI-549 binds with a characteristic propeller shape, as seen for multiple p110γ and p110δ selective inhibitors (*Berndt et al., 2010*). IPI-549 leads to a conformational change in the orientation of M804, which opens the specificity pocket, primarily composed of W812 and M804 (*Figure 5C*, *Figure 5—figure supplement 1D*). Comparison of IPI-549 bound to p110γ to the selective inhibitor Idelalisib bound to p110δ revealed a potential molecular mechanism for p110γ selectivity. Structure activity analysis of IPI-549 and its derivatives showed a critical role for substitutions at the alkyne position in achieving p110γ specificity (*Evans et al., 2016*). The *N*-methylpyrazole group in IPI-549 projects out of the specificity pocket towards the kα1-kα2 loop. This loop is significantly shorter in

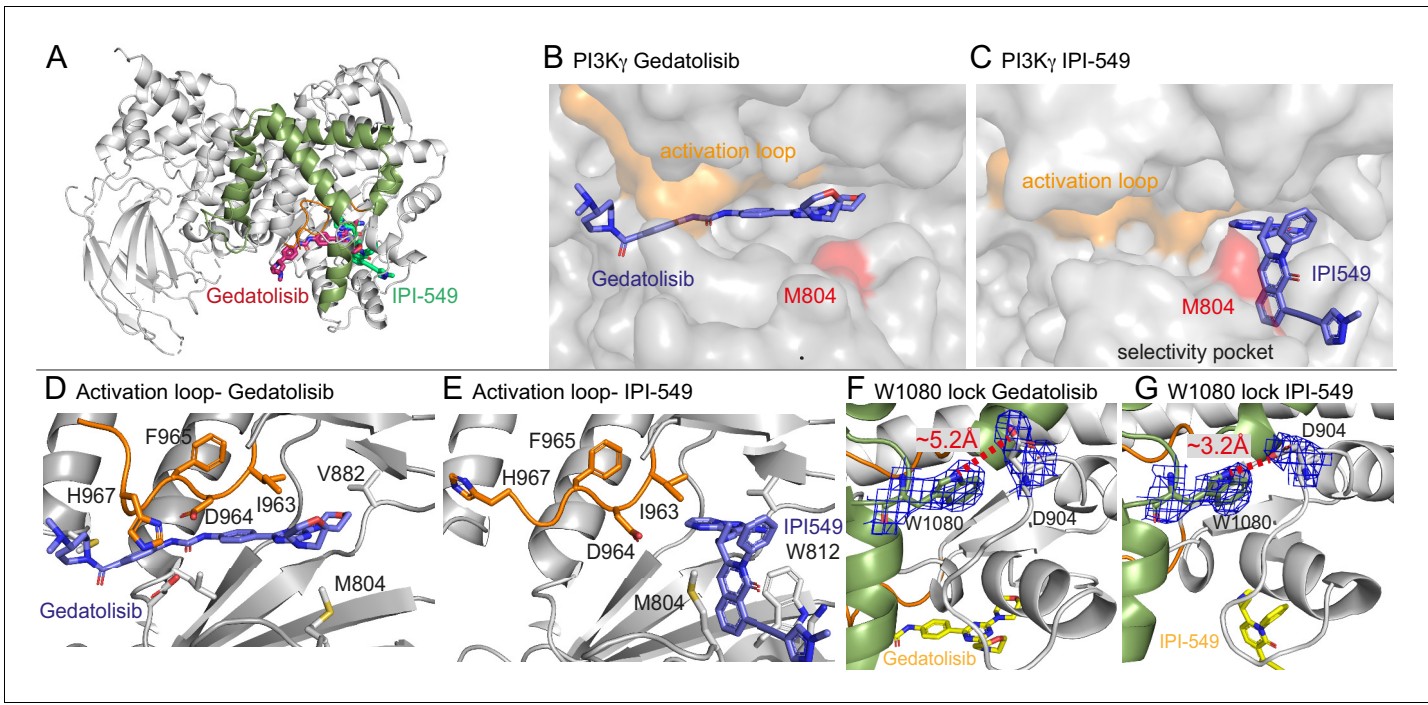

**Figure 5.** Structures of Gedatolisib and IPI-549 bound to p110γ. (**A**) Overall structure of Gedatolisib (red) and IPI-549 (green) bound to p110γ. (**B–C**) Comparison of Gedatolisib and IPI-549 bound to p110γ with the activation loop and selectivity pocket highlighted. M804 that changes conformation upon selectivity pocket opening is coloured red. (**D–E**) Comparison of the conformation of the activation loop (orange) of p110γ when Gedatolisib or IPI-549 are bound, with residues in the activation loop labelled, specifically D964 and F965 of the DFG motif labelled. (**F–G**) The Trp lock composed of W0180 is partially disrupted in the Gedatolisib structure compared to the IPI-549 structure. The interaction between W1080 and D904 is shown, with the distance between the two shown on each structure. The electron density from a feature enhanced map (*Afonine et al., 2015*) around W1080 and D904 in each structure is contoured at 1.5 sigma.

The online version of this article includes the following figure supplement(s) for figure 5:

**Figure supplement 1.** Structures of Gedatolisib, IPI-549, and NVS-PI3-4 bound to p110γ.

**Figure supplement 2.** Binding of IPI-549, NVS-PI3-4, and Gedatolisib lead to different conformations of the activation loop of p110γ.

p110δ, with a potential clash with bulkier alkyne derivatives (*Figure 5—figure supplement 1K+L*). However, this cannot be the main driver of specificity, as a phenyl substituent of the alkyne had decreased selectivity for p110γ over p110δ, with hydrophilic heterocycles in this position being critical in p110γ selectivity (*Evans et al., 2016*). A major difference in this pocket between p110γ and p110δ is K802 in p110γ (T750 in p110δ), with this residue making a pi-stacking interaction with W812. The *N*-methylpyrazole group packs against K802, with a bulkier group in this position likely to disrupt the pi stacking interaction, explaining the decreased potency for these compounds (*Evans et al., 2016*).

One of the most striking differences between the structure of Gedatolisib and IPI-549 bound to p110γ is the conformation of the N-terminus of the activation loop, including the residues that make up the DFG motif (*Figure 5B–E*, *Figure 5—figure supplement 2B–F*). The majority of the activation loop is disordered in PI3Kγ crystal structures, with the last residue being between 967 and 969. Gedatolisib makes extensive contacts with the activation loop, with H967 immediately following the DFG motif in a completely altered conformation. The interaction of the cyclopropyl motif in AZ2 with the activation loop has previously been proposed to be critical in mediating allosteric conformational changes. In addition to the change in the activation loop, there was a minor perturbation of the W1080 lock, with the Gedatolisib structure revealing a disruption of the hydrogen bond between W1080 and D904, with this bond maintained in the IPI-549 structure (*Figure 5F+G*). The C-terminus of the activation loop and kα7 immediately following showed some of the largest changes upon inhibitor binding in HDX experiments. The kα7 helix is directly in contact with W1080, and we postulated that the conformational changes induced in the N-terminus of the activation loop may mediate these changes.

## Conformational selective inhibitors show altered specificity towards activating PI3Kγ mutant

We observed that HDX differences occurring in the R1021C mutant were very similar to conformational changes observed for p110γ bound to Gedatolisib. This suggests that both the mutant and Gedatolisib bound forms might be adopting an activated conformation of the regulatory motif. HDX differences were very similar for the region spanning 976–992 in the activation loop (*Figure 2G*, *Figure 4E*). As this region is directly adjacent to the inhibitor binding site, we postulated that there may be altered inhibitor binding for the R1021C mutant. We carried out $IC_{50}$ measurement for wild-type and R1021C p110γ/p101 with both IPI-549 and Gedatolisib (*Figure 6A*). Gedatolisib was roughly threefold more potent for the R1021C mutant over the wild-type, with no significant difference in $IC_{50}$ values for IPI-549 compared to wild-type. This provides initial insights into how understanding the dynamics of activating mutations and inhibitors may be useful as a novel strategy towards designing mutant specific inhibitors.

## Discussion

Understanding the molecular determinants of how mutations in PI3Ks lead to altered signalling in disease is vital in the design of targeted therapeutic strategies. The PI3Kγ isoform is critical to maintain immune system function, and plays important roles in the regulation of the tumour microenvironment (*Fruman et al., 2017*; *Okkenhaug, 2013*). Bi-allelic loss-of-function mutations in p110γ are a driver of human immunodeficiencies, and multiple inactivating mutations located in the C-terminal regulatory motif of the kinase domain have been described (*Takeda et al., 2019*; *Thian et al., 2020*). Initial results linking deletion of PI3Kγ to the development of colon cancer (*Sasaki et al., 2000b*) have been disputed (*Barbier et al., 2001*), and recent studies suggest that tumour growth and metastasis is attenuated in PI3Kγ deficient mice (*Kaneda et al., 2016b*; *Torres et al., 2019*) and IPI-549-treated animals (*De Henau et al., 2016*). Inhibiting PI3Kγ has shown promise as an immunomodulatory agent in generating an antitumour immune response (*De Henau et al., 2016*; *Kaneda et al., 2016b*). There have also been numerous reports of overexpression and single-nucleotide variants in *PIK3CG* linked to cancer development in multiple tissues (*Torres et al., 2019*; *Dituri et al., 2012*; *Edling et al., 2010*; *Ge et al., 2019*; *Zhang et al., 2019*; *Nava Rodrigues et al., 2018*; *Shu et al., 2018*; *Wang et al., 2020*). Oncogenic mutations in *PIK3CG* are widely distributed, which is distinct from the oncogenic hotspot mutations seen in the helical and kinase domain of *PIK3CA*. There has been limited analysis of the functional consequences of oncogenic *PIK3CG*

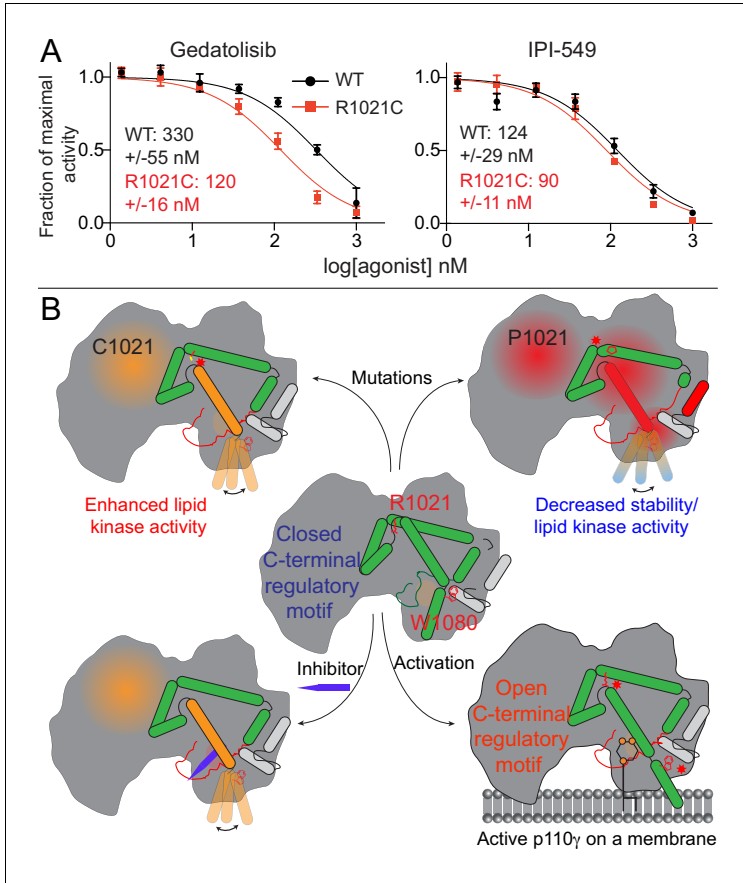

**Figure 6.** Activating mutations show slight differences in inhibition by allosteric inhibitors and model of PI3Kγ regulation. (**A**) $IC_{50}$ curves for wild-type and R1021C p110γ/p101 complexes. Assays were carried out with 5% C8 $PIP_2$/95% PS vesicles at a final concentration of 1 mg/ml in the presence of 100 μM ATP and 1.5 μM lipidated Gβγ. PI3Kγ concentration was 4 nM for R1021C and 8 nM for WT. Error is shown as standard deviation (n = 3). Refer to the *Figure 6—source data 1* file for $IC_{50}$ curve data. (**B**) Model of conformational changes that occur upon mutation of the C-terminal motif and binding of activation loop interacting conformation selective inhibitors. The online version of this article includes the following source data for figure 6:

**Source data 1.** Source data for IC50 curves comparing inhibition of WT and R1021C p110g/p101.

mutants, with the R1021 residue in the regulatory motif of the kinase domain being unique, as mutations of this residue exist in both immunodeficiencies and tumours.

Here, we have described the biochemical and biophysical characterisation of both activating and inactivating disease-linked R1021 mutations in the regulatory motif of the p110γ kinase domain. This has revealed that mutation of R1021 can lead to either kinase activation or inactivation. The R1021 in the kα8 helix is conserved across all class I PI3Ks, with it making a number of hydrogen bonds with residues in kα10 and kα11. Both R1021P and R1021C would lead to disruption of the hydrogen bonds with kα10 and kα11, however R1021P would also lead to disruption of the kα8 helix due to the altered dynamics introduced by the proline residue. HDX-MS results were consistent with this hypothesis, with R1021P leading to large-scale conformational changes across the entire protein, with the main disruptions occurring in the helical and kinase domain. The R1021P mutation greatly destabilised the protein, with purification yields being >20-fold lower than wild-type, consistent with greatly decreased p110γ and p101 levels in patient T cells (*Takeda et al., 2019*). Consistent with previous reports, we found greatly decreased lipid kinase activity for R1021P, although the enzyme maintained catalytic ability, as it showed greatly increased basal ATPase activity, which is similar to what occurs upon mutation of the W1080 lock or removal of the kα12 helix (*Gangadhara et al., 2019*; *Takeda et al., 2019*). Overall, this data is consistent with this mutation leading to

destabilisation of the p110γ protein, leading to both greatly decreased protein levels, and decreased lipid kinase activity.

The R1021C mutation, in contrast, had enhanced lipid kinase activity, both basally, and upon Gβγ activation. HDX-MS and MD simulations suggest that this activation is mediated by a conformational disruption of the inhibitory c-terminal regulatory motif of the kinase domain. Many of these changes in the C-terminal regulatory motif have been previously observed upon membrane binding (*Vadas et al., 2013*), as well as upon binding to conformational selective inhibitors (*Gangadhara et al., 2019*). One of the largest changes in exchange occurred at the C-terminus of the activation loop and the beginning of kα7 which is in contact with the W1080 lock. We propose a model of how mutation of R1021 can lead to either activated or inactivated lipid kinase activity (*Figure 6B*). The conformation of the C-terminal regulatory motif is critical in regulating lipid kinase activity, where minor perturbations (R1021C) can lead to disruption of multiple inhibitory contacts allowing for reorientation of the kα12 membrane binding helix and increased lipid kinase activity. This suggests that the mutation can pre-organise the enzyme for productive catalysis, either through increased membrane binding, or increased $PIP_2$ kinase activity. Reinforcing the importance of the R1021 mutant for class I PI3K regulation is the fact that mutation of the equivalent R992 in *PIK3CA* to either Leu or Asn has been found in tumour samples (*Tate et al., 2019*).

This work corroborates the important role of the C-terminal regulatory motif in controlling PI3K lipid kinase activity. The orientation of this motif is critical in the regulation of all class I PI3Ks, although this is regulated by different molecular mechanisms in p110α, p110β, p110δ, and p110γ. The class IA PI3Ks require p85 regulatory subunits to stabilise the C-terminal regulatory motif, with the nSH2 of p85 interacting with and stabilising kα10 for all class IA PI3Ks (*Burke and Williams, 2013*; *Mandelker et al., 2009*), and the cSH2 of p85 stabilising kα7, kα8, kα11, and kα12 for p110β and p110δ (*Burke et al., 2011*; *Zhang et al., 2011*). The p110γ isoform is unique in that its C-terminal motif adopts an inhibited conformation in the absence of regulatory proteins. The C-terminal regulatory motif of p110γ can be post-translationally modified by phosphorylation of kα9 (T1024) by protein kinase A decreasing lipid kinase activity (*Perino et al., 2011*), while protein kinase C phosphorylates an adjacent area in the helical domain (S582) (*Walser et al., 2013*) increasing lipid kinase activity.

Extensive work has been carried out toward the development of allosteric inhibitors of PI3Ks. Although no allosteric inhibitors that bind out of the ATP-binding pocket have been developed, fragment screens have revealed potential druggable pockets that may be targeted in class IA PI3Ks (*Miller et al., 2017*). It has previously been noted that PI3Kγ can be selectively targeted through a conformationally selective inhibitor, AZ2 (*Gangadhara et al., 2019*). This was mediated through a cyclopropyl moiety on AZ2, which putatively alters the orientation of the activation loop, leading to disruption of the inhibitory conformation of the C-terminal regulatory motif. Many of the changes observed for this inhibitor were similar to those seen in the R1021C mutant. To interrogate if allosteric conformational changes were unique to cyclopropyl containing compounds, we screened a panel of pan-PI3K and PI3Kγ selective inhibitors using HDX-MS. The compounds PIK90, IPI549, and AS-604850 only caused decreased exchange at the active site. Comparison of the crystal structures of these compounds (*Camps et al., 2005*; *Knight et al., 2006*) revealed similar conformations of the activation loop, with limited interaction between the inhibitors and the activation loop. After submission of this manuscript, the structure of IPI-549 was released by Arcus Biosciences (*Drew et al., 2020*) with a conserved binding mode to the model proposed here.

The AZ2 compound containing the cyclopropyl moiety, led to large-scale conformational changes consistent with previous results (*Gangadhara et al., 2019*). Intriguingly, the non-specific inhibitors, Gedatolisib and Omipalisib, caused increased exchange in many of the same regions that showed enhanced exchange with the R1021C mutant. Comparison of the crystal structures of these inhibitors revealed more extensive interactions with the activation loop and significant conformational rearrangement of the activation loop. Distinct from the AZ2 compound, neither Gedatolisib and Omipalisib show specificity for PI3Kγ over class IA PI3Ks (*Knight et al., 2010*; *Venkatesan et al., 2010*). Similar HDX-MS differences were observed for both the R1021C mutant and wild type bound to Gedatolisib. Gedatolisib showed increased potency versus R1021C over wild-type PI3Kγ, with a ~ 3 fold decrease in $IC_{50}$ values. Altogether, this suggests that R1021C induces a conformation similar to the wild type enzyme bound to Gedatolisib. This provides an intriguing approach for designing

oncogenic PI3K-specific inhibitors through further optimisation of the ATP competitive inhibitor moieties in the activation loop binding region.

Overall, this work provides novel insights into how the C-terminal regulatory motif of PI3Kγ regulates lipid kinase activity, how oncogenic and immunodeficiency mutations can disrupt this regulation, and how one can exploit these conformational changes to develop isoform and mutant selective small molecule inhibitors. Further exploration of the dynamic regulation of the C-terminal regulatory motif of PI3Ks by mutations and inhibitors may reveal unique approaches to develop therapeutics for PI3K-related human diseases.

# Materials and methods

## Key resources table

| Reagent type (species) or resource | Designation | Source or reference | Identifiers | Additional information |
|---|---|---|---|---|
| Cell Line (*Spodoptera frugiperda*) | Sf9 | Expression Systems | 94–001S | Insect cell line for protein expression |
| Recombinant DNA reagent | pACEBac1-p110γ (plasmid) | This paper | MR30 | PIK3CG Sf9 expression vector (available from Burke lab) |
| Recombinant DNA reagent | pACEBac1-p110γ R1021C (plasmid) | This paper | MR53 | PIK3CG Sf9 expression vector (available from Burke lab) |
| Recombinant DNA reagent | pMultiBac-p110γ/ p101(plasmid) | This paper | MR22 | PIK3CG/PIK3R5 Sf9 expression vector (available from Burke lab) |
| Recombinant DNA reagent | pMultiBac-p110γ R1021C /p101(plasmid) | This paper | JS39 | PIK3CG/PIK3R5 Sf9 expression vector (available from Burke lab) |
| Recombinant DNA reagent | pbiGBac-p110γ R1021P /p101(plasmid) | This paper | MR92 | PIK3CG/PIK3R5 Sf9 expression vector (available from Burke lab) |
| Recombinant DNA reagent | pFastBac-HRas G12V (plasmid) | This paper | BS9 | HRAS Sf9 expression vector (available from Burke lab) |
| Recombinant DNA reagent | pMultiBac-Gβ1/Gγ2 (plasmid) | Oscar Vadas | pOP737 | GBB1/GG2 Sf9 expression vector (available from Burke lab) |
| Recombinant DNA reagent | pACEBac1-p110γ (144–1102) (plasmid) | This paper | MR7 | PIK3CG Sf9 expression vector (available from Burke lab) |
| Sequence-based reagent | Fwd primer for R1021C mutation | Sigma Aldrich | PCR primers | GGCTTATCTAGCCCTTTGTCATCA CACAAACCTACTGATCATCCTGTTC |
| Sequence-based reagent | Rev primer for R1021C mutation | Sigma Aldrich | PCR primers | AAGGGCTAGATAAGCC TTAACACAGATG |
| Sequence-based reagent | Fwd primer for R1021P mutation | Sigma Aldrich | PCR primers | CCTTCCTCATCACACAAACC TACTGATCATCCTGTTCTCC |
| Sequence-based reagent | Rev primer for R1021P mutation | Sigma Aldrich | PCR primers | GATGAGGAAGGGCTAGATAA GCCTTAACACAGATGTCCTG |
| Commercial assay or kit | Transcreener ADP2 FI assay (1,000 Assay, 384 Well) | BellBrook Labs | 3013–1K | Lipid Kinase activity assay kit |
| Chemical compound, drug | GTPγS | Sigma Aldrich | G8634 | GTPγS for HRas loading |
| Chemical compound, drug | $D_2O$ | Sigma Aldrich | 151882 | Heavy water for HDX |
| Chemical compound, drug | L-α-Phosphatdiylcholine | Avanti | 840051C | |
| Chemical compound, drug | L-α-Phosphatidylethanolamine | Sigma Aldrich | P6386 | |
| Chemical compound, drug | L-α-Phosphatidylserine | Avanti | 840032C | |
| Chemical compound, drug | L-α-phosphatidylinositol-4,5-bisphosphate | Avanti | 840046X | |

*Continued on next page*

*Continued*

| Reagent type (species) or resource | Designation | Source or reference | Identifiers | Additional information |
|---|---|---|---|---|
| Chemical compound, drug | diC8 phosphatidylinositol-4,5-bisphosphate | Echelon Biosciences | P-4508 | |
| Chemical compound, drug | Cholesterol | Sigma Aldrich | 47127 U | |
| Chemical compound, drug | Sphingomyelin | Sigma Aldrich | S0756 | |
| Chemical compound, drug | IPI-549 | Chemie Tex | CT-IPI549 | PMID:27660692 |
| Chemical compound, drug | PIK-90 | Axon Medchem | Axon1362 | PMID:19318683 |
| Chemical compound, drug | AS-604850 | Sigma Aldrich | 528108 | PMID:16127437 |
| Chemical compound, drug | Gedatolisib PF-05212384 PKI587 | Bionet | FE-0013 | PMID:20166697 |
| Chemical compound, drug | Omipalisib (GSK2126458, GSK458) | LuBio Science | S2658 | PMID:24900173 |
| Chemical compound, drug | NVS-PI3-4 | Haouyan Chemexpress | HY-133907 | PMID:22863202 |
| Chemical compound, drug | AZ2 | Haouyan Chemexpress | HY-111570 | PMID:30718815 |
| Software, algorithm | HDExaminer | Sierra Analytics | http://massspec.com/hdexaminer | |
| Software, algorithm | GraphPad Prism 7 | GraphPad | https://www.graphpad.com/scientific-software/prism/ | |
| Software, algorithm | PyMOL | Schroedinger | http://pymol.org | |

## Expression and purification of PI3Kγ constructs

Full-length monomeric p110γ (WT, R1021C) and p110γ/p101 complex (WT, R1021C, R1021P) were expressed in Sf9 insect cells using the baculovirus expression system. For the complex, the subunits were co-expressed from a MultiBac vector (*Berger et al., 2004*). Following 55 hr of expression, cells were harvested by centrifuging at 1680 RCF (Eppendorf Centrifuge 5810 R) and the pellets were snap-frozen in liquid nitrogen. Both the monomer and the complex were purified identically through a combination of nickel affinity, streptavidin affinity and size exclusion chromatographic techniques.

Frozen insect cell pellets were resuspended in lysis buffer (20 mM Tris pH 8.0, 100 mM NaCl, 10 mM imidazole pH 8.0, 5% glycerol [v/v], 2 mM beta-mercaptoethanol [βME], protease inhibitor [Protease Inhibitor Cocktail Set III, Sigma]) and sonicated for 2 min (15 s on, 15 s off, level 4.0, Misonix sonicator 3000). Triton-X was added to the lysate to a final concentration of 0.1% and clarified by spinning at 15,000 g for 45 min (Beckman Coulter JA-20 rotor). The supernatant was loaded onto a 5 mL HisTrap FF crude column (GE Healthcare) equilibrated in NiNTA A buffer (20 mM Tris pH 8.0, 100 mM NaCl, 20 mM imidazole pH 8.0, 5% [v/v] glycerol, 2 mM βME). The column was washed with high-salt NiNTA A buffer (20 mM Tris pH 8.0, 1 M NaCl, 20 mM imidazole pH 8.0, 5% [v/v] glycerol, 2 mM βME), NiNTA A buffer, 6% NiNTA B buffer (20 mM Tris pH 8.0, 100 mM NaCl, 250 mM imidazole pH 8.0, 5% [v/v] glycerol, 2 mM βME) and the protein was eluted with 100% NiNTA B. The eluent was loaded onto a 5 mL StrepTrap HP column (GE Healthcare) equilibrated in gel filtration buffer (20 mM Tris pH 8.5, 100 mM NaCl, 50 mM Ammonium Sulfate and 0.5 mM tris(2-carboxyethyl) phosphine [TCEP]). The column was washed with the same buffer and loaded with tobacco etch virus protease. After cleavage on the column overnight, the protein was eluted in gel filtration buffer. The eluent was concentrated in a 50,000 MWCO Amicon Concentrator (Millipore) to <1 mL and injected onto a Superdex 200 10/300 GL Increase size-exclusion column (GE Healthcare) equilibrated in gel filtration buffer. After size exclusion, the protein was concentrated, aliquoted, frozen, and stored at −80°C.

For crystallography, p110γ (144–1102) was expressed in Sf9 insect cells for 72 hr. The cell pellet was lysed, and the lysate was subjected to nickel affinity purification as described above. The eluent was loaded onto HiTrap Heparin HP cation exchange column equilibrated in Hep A buffer (20 mM Tris pH 8.0, 100 mM NaCl, 5% glycerol and 2 mM βME). A gradient was started with Hep B buffer (20 mM Tris pH 8.0, 1 M NaCl, 5% glycerol and 2 mM βME) and the fractions containing the peak were pooled. This was then loaded onto HiTrap Q HP anion exchange column equilibrated with Hep A and again subjected to a gradient with Hep B. The peak fractions were pooled, concentrated on a 50,000 MWCO Amicon Concentrator (Millipore) to <1 mL and injected onto a Superdex 200 10/300 GL Increase size-exclusion column (GE Healthcare) equilibrated in gel filtration buffer (20 mM Tris pH 7.2, 0.5 mM $(NH_4)_2SO_4$, 1% ethylene glycol, 0.02% CHAPS, and 5 mM DTT). Protein from size exclusion was concentrated to >5 mg/mL, aliquoted, frozen and stored at −80˚C.

## Expression and purification of lipidated Gβγ

Full-length, lipidated Gβγ was expressed in Sf9 insect cells and purified as described previously (*Kozasa, 2004*). After 65 hr of expression, cells were harvested and the pellets were frozen as described above. Pellets were resuspended in lysis buffer (20 mM HEPES pH 7.7, 100 mM NaCl, 10 mM βME, protease inhibitor [Protease Inhibitor Cocktail Set III, Sigma]) and sonicated for 2 min (15 s on, 15 s off, level 4.0, Misonix sonicator 3000). The lysate was spun at 500 RCF (Eppendorf Centrifuge 5810 R) to remove intact cells and the supernatant was centrifuged again at 25,000 g for 1 hr (Beckman Coulter JA-20 rotor). The pellet was resuspended in lysis buffer and sodium cholate was added to a final concentration of 1% and stirred at 4˚C for 1 hr. The membrane extract was clarified by spinning at 10,000 g for 30 min (Beckman Coulter JA-20 rotor). The supernatant was diluted three times with NiNTA A buffer (20 mM HEPES pH 7.7, 100 mM NaCl, 10 mM Imidazole, 0.1% C12E10, 10 mM βME) and loaded onto a 5 mL HisTrap FF crude column (GE Healthcare) equilibrated in the same buffer. The column was washed with NiNTA A, 6% NiNTA B buffer (20 mM HEPES pH 7.7, 25 mM NaCl, 250 mM imidazole pH 8.0, 0.1% C12E10, 10 mM βME) and the protein was eluted with 100% NiNTA B. The eluent was loaded onto HiTrap Q HP anion exchange column equilibrated in Hep A buffer (20 mM Tris pH 8.0, 8 mM CHAPS, 2 mM Dithiothreitol [DTT]). A gradient was started with Hep B buffer (20 mM Tris pH 8.0, 500 mM NaCl, 8 mM CHAPS, 2 mM DTT) and the protein was eluted in ~50% Hep B buffer. The eluent was concentrated in a 30,000 MWCO Amicon Concentrator (Millipore) to <1 mL and injected onto a Superdex 75 10/300 GL size exclusion column (GE Healthcare) equilibrated in Gel Filtration buffer (20 mM HEPES pH 7.7, 100 mM NaCl, 10 mM CHAPS, 2 mM TCEP). Fractions containing protein were pooled, concentrated, aliquoted, frozen and stored at −80˚C.

## Expression and purification of lipidated HRas G12V

Full-length HRas G12V was expressed by infecting 500 mL of Sf9 cells with 5 mL of baculovirus. Cells were harvested after 55 hr of infection and frozen as described above. The frozen cell pellet was resuspended in lysis buffer (50 mM HEPES pH 7.5, 100 mM NaCl, 10 mM βME and protease inhibitor (Protease Inhibitor Cocktail Set III, Sigma)) and sonicated on ice for 1 min 30 s (15 s ON, 15 s OFF, power level 4.0) on the Misonix sonicator 3000. Triton-X 114 was added to the lysate to a final concentration of 1%, mixed for 10 min at 4˚C and centrifuged at 25,000 rpm for 45 min (Beckman Ti-45 rotor). The supernatant was warmed to 37˚C for few minutes until it turned cloudy following which it was centrifuged at 11,000 rpm at room temperature for 10 min (Beckman JA-20 rotor) to separate the soluble and detergent-enriched phases. The soluble phase was removed, and Triton-X 114 was added to the detergent-enriched phase to a final concentration of 1%. Phase separation was performed three times. Imidazole pH 8.0 was added to the detergent phase to a final concentration of 15 mM and the mixture was incubated with Ni-NTA agarose beads (Qiagen) for 1 hr at 4˚C. The beads were washed with five column volumes of Ras-NiNTA buffer A (20 mM Tris pH 8.0, 100 mM NaCl, 15 mM imidazole pH 8.0, 10 mM βME and 0.5% Sodium Cholate) and the protein was eluted with two column volumes of Ras-NiNTA buffer B (20 mM Tris pH 8.0, 100 mM NaCl, 250 mM imidazole pH 8.0, 10 mM βME and 0.5% Sodium Cholate). The protein was buffer exchanged to Ras-NiNTA buffer A using a 10,000 kDa MWCO Amicon concentrator, where protein was concentrated to ~1 mL and topped up to 15 mL with Ras-NiNTA buffer A and this was repeated a total of three times. GTPγS was added in twofold molar excess relative to HRas along with 25 mM EDTA. After

incubating for an hour at room temperature, the protein was buffer exchanged with phosphatase buffer (32 mM Tris pH 8.0, 200 mM Ammonium Sulphate, 0.1 mM $ZnCl_2$, 10 mM βME and 0.5% Sodium Cholate). One unit of immobilised calf alkaline phosphatase (Sigma) was added per milligram of HRas along with twofold excess nucleotide and the mixture was incubated for 1 hr on ice. $MgCl_2$ was added to a final concentration of 30 mM to lock the bound nucleotide. The immobilised phosphatase was removed using a 0.22-micron spin filter (EMD Millipore). The protein was concentrated to less than 1 mL and was injected onto a Superdex 75 10/300 GL size exclusion column (GE Healthcare) equilibrated in gel filtration buffer (20 mM HEPES pH 7.7, 100 mM NaCl, 10 mM CHAPS, 1 mM $MgCl_2$ and 2 mM TCEP). The protein was concentrated to 1 mg/mL using a 10,000 kDa MWCO Amicon concentrator, aliquoted, snap-frozen in liquid nitrogen and stored at −80°C.

## Lipid vesicle preparation

For kinase assays comparing WT and mutant activities, lipid vesicles containing 5% brain phosphatidylinositol 4,5- bisphosphate (PI*P2*), 20% brain phosphatidylserine (PS), 50% egg-yolk phosphatidylethanolamine (PE), 10% egg-yolk phosphatidylcholine (PC), 10% cholesterol and 5% egg-yolk sphingomyelin (SM) were prepared by mixing the lipids dissolved in organic solvent. The solvent was evaporated in a stream of argon following which the lipid film was desiccated in a vacuum for 45 min. The lipids were resuspended in lipid buffer (20 mM HEPES pH 7.0, 100 mM NaCl and 10% glycerol) and the solution was sonicated for 15 min. The vesicles were subjected to five freeze thaw cycles and extruded 11 times through a 100 nm filter (T and T Scientific: TT-002–0010). The extruded vesicles were sonicated again for 5 min, aliquoted and stored at −80°C. For inhibitor response assays, lipid vesicles containing 95% PS and 5% C8-PI*P2* were used. PS was dried and desiccated as described above. The lipid film was mixed and resuspended with C8-PI*P2* solution (2.5 mg/mL in lipid buffer). Following this, vesicles were essentially prepared the same way as described above. All vesicles were stored at 5 mg/mL.

## Lipid kinase assays

All lipid kinase activity assays employed the Transcreener ADP2 Fluorescence Intensity (FI) Assay (Bellbrook labs) which measures ADP production. For assays comparing the activities of mutants, final concentrations of PM-mimic vesicles were 1 mg/mL, ATP was 100 µM ATP and lipidated Gβγ/HRas were at 1.5 µM. Two µL of a PI3K solution at 2X final concentration (50–3000 nM final) was mixed with 2 µL substrate solution containing ATP, vesicles and Gβγ/HRas or Gβγ/HRas gel filtration buffer and the reaction was allowed to proceed for 60 min at 20°C. The reaction was stopped with 4 µL of 2X stop and detect solution containing Stop and Detect buffer, 8 nM ADP Alexa Fluor 594 Tracer and 93.7 µg/mL ADP2 Antibody IRDye QC-1 and incubated for 50 min. The fluorescence intensity was measured using a SpectraMax M5 plate reader at excitation 590 nm and emission 620 nm. This data was normalised against a 0–100% ADP window made using conditions containing either 100 µM ATP/ADP with vesicles and kinase buffer. % ATP turnover was interpolated from an ATP standard curve obtained from performing the assay on 100 µM (total) ATP/ADP mixtures with increasing concentrations of ADP using Prism 7. All specific activities of lipid kinase activity were corrected for the basal ATPase activity by subtracting the specific activity of the WT/mutant protein in the absence of vesicles/activators.

For assays measuring inhibitor response, substrate solutions containing vesicles, ATP and Gβγ at 4X final concentration (as described above) were mixed with 4X solutions of inhibitor dissolved in lipid buffer (<1% DMSO) in serial to obtain 2X substrate solutions with inhibitors at the various 2X concentrations. Two µL of this solution was mixed with 2 µL of 2X protein solution (4 nM final for the mutant and 8 nM final for WT) to start the reaction and allowed to proceed for 60 min at 37°C. Following this, the reaction was stopped and the intensity was measured. The raw data was normalised against a 0–100% ADP window as described above. The % inhibition was calculated by comparison to the activity with no inhibitor to obtain fraction activity remaining.

## Hydrogen deuterium exchange mass spectrometry (HDX-MS)

HDX experiments were performed similarly as described before (*Dornan et al., 2017*). For HDX with mutants, 3 µL containing 13 picomoles of protein was incubated with 8.25 µL of $D_2O$ buffer (20 mM HEPES pH 7.5, 100 mM NaCl, 98% (v/v) $D_2O$) for four different time periods (3, 30, 300, 3000 s at

20°C). After the appropriate time, the reaction was stopped with 57.5 µL of ice-cold quench buffer (2M guanidine, 3% formic acid), immediately snap frozen in liquid nitrogen and stored at −80°C. For HDX with inhibitors, 5 µL of p110γ or p110γ/p101 at 2 µM was mixed with 5 µL of inhibitor at 4 µM in 10% DMSO or 5 µL of blank solution containing 10% DMSO and incubated for 20 min on ice. A total of 40 µL of D2O buffer was added to this solution to start the exchange reaction which was allowed to proceed for four different time periods (3, 30, 300, 3000 s at 20°C). After the appropriate time, the reaction was terminated with 20 µL of ice-cold quench buffer and the samples were frozen.

Protein samples were rapidly thawed and injected onto an ultra-high-pressure liquid chromatography (UPLC) system at 2°C. Protein was run over two immobilised pepsin columns (Trajan, ProDx protease column, PDX.PP01-F32 and Applied Biosystems, Porosyme, 2-3131-00) at 10°C and 2°C at 200 µL/min for 3 min, and peptides were collected onto a VanGuard precolumn trap (Waters). The trap was subsequently eluted in line with an Acquity 1.7 µm particle, $100 \times 1$ mm$^2$ C18 UPLC column (Waters), using a gradient of 5–36% B (buffer A, 0.1% formic acid; buffer B, 100% acetonitrile) over 16 min. Mass spectrometry experiments were performed on an Impact II TOF (Bruker) acquiring over a mass range from 150 to 2200 m/z using an electrospray ionisation source operated at a temperature of 200°C and a spray voltage of 4.5 kV. Peptides were identified using data-dependent acquisition methods following tandem MS/MS experiments (0.5 s precursor scan from 150 to 2000 m/z; 12 0.25 s fragment scans from 150 to 2000 m/z). MS/MS datasets were analysed using PEAKS7 (PEAKS), and a false discovery rate was set at 1% using a database of purified proteins and known contaminants.

HD-Examiner software (Sierra Analytics) was used to automatically calculate the level of deuterium incorporation into each peptide. All peptides were manually inspected for correct charge state and presence of overlapping peptides. Deuteration levels were calculated using the centroid of the experimental isotope clusters. The results for these proteins are presented as the raw percent deuterium incorporation, as shown in Supplemental Data, with the only correction being applied correcting for the deuterium oxide percentage of the buffer utilised in the exchange (62% for experiments with mutants and 75.5% for experiments with inhibitors). No corrections for back exchange that occurs during the quench and digest/separation were applied. Attempts to generate a fully deuterated class I PI3K sample were unsuccessful, which is common for large macromolecular complexes. Therefore, all deuterium exchange values are relative.

Changes in any peptide at any time point greater than both 5% and 0.4 Da between conditions with a paired t test value of p<0.01 were considered significant. Peptides that crossed these criteria were mapped onto the structures in *Figure 2A+B* and *Figure 4B-D*. Peptides that crossed this threshold are annotated on the raw HDX data in the source data excel file according to the legend. The raw HDX data are shown in two different formats. The raw peptide deuterium incorporation graphs for a selection of peptides with significant differences are shown, with the raw data for all analysed peptides in the source data. To allow for visualisation of differences across all peptides, we utilised number of deuteron difference (#D) plots. These plots show the total difference in deuterium incorporation over the entire H/D exchange time course, with each point indicating a single peptide. These graphs are calculated by summing the differences at every time point for each peptide and propagating the error (example *Figure 2C–F*). This visualisation was utilised over the similar strategy of graphing differences at each timepoint separately as it allowed for the display of multiple comparisons on the same graph (example *Figure 4A*).

The mass spectrometry proteomics data have been deposited to the ProteomeXchange Consortium via the PRIDE partner repository (*Perez-Riverol et al., 2019*) with the dataset identifier PXD021132.

## X-ray crystallography:

p110γ (144–1102) was crystallised from a grid of 2 µL sitting drops at 1:1, 2:1 and 3:1 protein to reservoir ratios at 18°C. Protein at 4 mg/mL (in 20 mM Tris pH 7.2, 0.5 mM (NH$_4$)$_2$SO$_4$, 1% ethylene glycol, 0.02% CHAPS and 5 mM DTT) was mixed with reservoir solution containing 100 mM Tris pH 7.5, 250 mM (NH$_4$)$_2$SO$_4$ and 20–22% PEG 4000. Large multinucleate crystals were generated in these drops. Inhibitor stocks were prepared at concentrations of 0.01 mM, 0.1 mM and 1 mM in cryo-protectant solution containing 100 mM Tris pH 7.5, 250 mM (NH$_4$)$_2$SO$_4$, 23% PEG 4000% and 14% glycerol. Inhibitors at increasing concentrations were added to the drops stepwise every 1 hr. After overnight incubation with the inhibitor, single crystals were manually broken from the multi-nucleates

and soaked in a fresh drop containing 1 mM inhibitor in cryo-protectant before being immediately frozen in liquid nitrogen. The final crystals were rod/needle shaped and had dimensions of roughly 10 × 20×80 µm, and were mounted on either 0.1–0.2 or 0.05–0.1 mm cryo loops (Hampton Research). Each unique structure of PI3K bound to inhibitors was solved from a single crystal.

Diffraction data for PI3Kγ crystals were collected on beamline 08ID-1 of the Canadian Light Source. Data was collected at 0.97949 Å using a beam width of 50 µm. Diffraction data for each unique crystal was collected using a strategy of 0.2–0.4 s exposure/0.2° rotation for each image, over a total rotational range of 180°. Data were processed using XDS (*Kabsch, 2010*). Phases were initially obtained by molecular replacement using Phaser (*McCoy et al., 2007*) using PDB: 2CHW for the IPI-549 complex (*Knight et al., 2006*), and 5JHA for Gedatolisib and NVS-PI3-4 (*Bohnacker et al., 2017*). Iterative model building and refinement were performed in COOT (*Emsley et al., 2010*) and phenix.refine (*Afonine et al., 2012*). Refinement was carried out with rigid body refinement, followed by translation/libration/screw B-factor and xyz refinement, with the final round of refinement optimising X-ray/stereochemistry and X-ray/ADP weighting. The final model was verified in Molprobity (*Chen et al., 2010*) to examine all Ramachandran and Rotamer outliers. Data collection and refinement statistics are shown in *Supplementary file 6*. The crystallography data has been deposited in the protein data bank with accession numbers (PDB: 7JWE, 7J × 0, 7JWZ).

## Molecular dynamics: missing loops modelling

The employed crystallographic structures of the p110γ protein reveal several missing gaps corresponding to flexible loops within range of the ligand-binding site: the activation loop (residues 968–981), and loops connecting the C2 and helical domains (residues 435–460 and 489–497). These missing gaps were modelled as disordered loops using Modeller9.19 (*Sali and Blundell, 1993*). Keeping the crystallographic coordinates fixed, 50 models were independently generated for each system. The wild type (WT), R1021C, and R1021P systems used PDB ID 6AUD (*Safina et al., 2017*) with their corresponding mutations in the mutant systems. The alignment used by Modeller between the crystallographic structure sequences and the FASTA sequence of p110γ (Uniprot ID P48736) were generated using Clustal Omega (*Sievers et al., 2011*). The top models were visually inspected to discard those in which loops were entangled in a knot or clashed with the rest of the structure. Lastly, from the remaining models, three were selected for each system to initiate simulations in triplicates.

## Molecular dynamics: system preparation

The generated models were prepared using tleap program of the AMBER package (*Case et al., 2005*). The systems were parametrised using the general AMBER force field (GAFF) using ff14sb for the protein (*Maier et al., 2015*). The systems were fully solvated with explicit water molecules described using the TIP3P model (*Jorgensen et al., 1983*), adding K+ and Cl- counterions to neutralise the total charge. The total number of atoms is 97,861 for WT (size: 116 Å ×95 Å × 94 Å), 100,079 for R1021C (size: 116 Å ×95 Å × 94 Å), 97,861 for R1021P (size: 116 Å ×95 Å × 94 Å).

## Gaussian accelerated molecular dynamics (GaMD)

All-atom MD simulations were conducted using the GPU version of AMBER18 (*Case et al., 2005*). The systems were initially relaxed through a series of minimisation, heating, and equilibration cycles. During the first cycle, the protein was restrained using a harmonic potential with a force constant of 10 kcal/mol-Å$^2$, while the solvent, and ions were subjected to an initial minimisation of 2000 steps using the steepest descent approach for 1000 steps and conjugate gradient approach for another 1000 steps. The full system (protein + solvent) was then similarly minimised for 1000 and 4000 steps using the steepest descent and conjugate gradient approaches, respectively. Subsequently, the temperature was incrementally changed from 100 to 300 K for 10 ps at two fs/step (NVT ensemble). Next, the systems were equilibrated for 200 ps at 1 atom and 300K (NPT ensemble), and for 200ps at 300K (NVT ensemble). Lastly, more equilibration simulations were run in the NVT ensemble in two steps; all systems were simulated using conventional MD for 50 ns and GaMD for 50ns. Temperature control (300 K) and pressure control (one atm) were performed via Langevin dynamics and Berendsen barostat, respectively. Production GaMD were simulated for ~3 µs for WT, ~4.1 µs R1021C, ~1.5 µs for R1021P. GaMD is an unconstrained enhanced sampling approach that works by adding a

harmonic boost potential to smooth biomolecular potential energy surface and reduce the system energy barriers (*Miao et al., 2015*). Details of the GaMD method have been extensively described in previous studies (*Miao et al., 2015*; *Pang et al., 2017*).

## GaMD analysis: principal component analysis (PCA)

PCA was performed using the sklearn.decomposition.PCA function in the *Scikit-learn* library using python3.6.9. First, all simulations were aligned with *mdtraj* (*McGibbon et al., 2015*) onto the same initial coordinates using Cα atoms of the kinase domain (residues 726–1088). Next, simulation coordinates of each domain of interest (for example kα9-kα10) from all systems (WT, R1021C, and R1021P) and replicas were concatenated and used to fit the transformation function. Subsequently, the fitted transformation function was applied to reduce the dimensionality of each domain's simulation Cα coordinates. It is important to note that all systems are transformed into the same PC space to evaluate the simulation variance across systems.

## GaMD analysis: angles calculation

Inter-helical angles were calculated using in-house python scripts along with *mdtraj* (*McGibbon et al., 2015*) as the angle between two vectors representing the principal axis along each helix. Each principal axis connects two points corresponding to the center of mass (COM) of the first and last turn from each helix. For kα8, points 1 and 2 are represented by the COM of residues 1020–1023 and 1004–1007 Cα coordinates, respectively. For kα9, points 1 and 2 are represented by the COM of residues 1024–1027 and 1034–1037 Cα coordinates, respectively. For kα10, points 1 and 2 are represented by the COM of residues 1053–1056 and 1046–1049 Cα coordinates, respectively. For kα11, points 1 and 2 are represented by the COM of residues 1062–1065 and 1074–1077 Cα coordinates, respectively. Angles were computed at each frame along the trajectories after structural alignment onto the initial coordinates using the Cα atoms of the kinase domain (residues 726–1088) as a reference.

## GaMD analysis: hydrogen bonds calculation

Hydrogen bonds were calculated using the *baker hubbard* command implemented with *mdtraj* (*McGibbon et al., 2015*). Occupancy (%) was determined by counting the number of frames in which a specific hydrogen bond was formed with respect to the total number of frames and then averaged across replicas.

## GaMD analysis: root-mean-square-fluctuations (RMSF)

RMSF was calculated using in-house python scripts along with *mdtraj* (*McGibbon et al., 2015*) RMSF was computed for each residue atom and represented as box plot to show the range of RMSF values across replicas. The trajectories were aligned onto the initial coordinates using the Cα atoms of the kinase domain (residues 726–1088) as a reference.

## PI3K inhibitors

Compounds were purchased from companies indicated below in ≥95% purity (typical 98% pure). IPI-549 (*Evans et al., 2016*) was purchased from ChemieTex (Indianapolis, USA, #CT-IPI549); PIK-90 (*Knight et al., 2006*) from Axon Medchem (Groningen, The Netherlands, #Axon1362); AS-604850 (PI 3-Kγ Inhibitor II, Calbiochem) (*Camps et al., 2005*) from Sigma Aldrich (#528108); Gedatolisib (PF-05212384, PKI587) (*Venkatesan et al., 2010*) from Bionet (Camelford, UK, #FE-0013); Omipalisib (GSK2126458, GSK458) (*Knight et al., 2010*) from LuBioScience GmbH (Zurich, Switzerland, #S2658); NVS-PI3-4 (*Collmann et al., 2013*; *Bruce et al., 2012*) and AZg1 (AZ2) (*Gangadhara et al., 2019*) from Haoyuan Chemexpress Co., Ltd. (Shanghai, China, #HY-133907 and #HY-111570, respectively).

## Acknowledgements

JEB is supported by a new investigator grant from the Canadian Institute of Health Research (CIHR), a Michael Smith Foundation for Health Research (MSFHR) Scholar award (17686), and an operating grant from the Cancer Research Society (CRS-24368). REA and ZG are supported in part by NIH

GM132826 and NCI P01-CA234228. MPW is funded by the Stiftung für Krebsbekämpfung grant 341, the Swiss National Science Foundation grant 310030_189065, the Novartis Foundation for medical-biological Research grant 14B095; and the Innosuisse grant 37213.1 IP-LS. We appreciate feedback from members of the Burke lab during preparation.

## Additional information

### Funding

| Funder | Grant reference number | Author |
|---|---|---|
| Cancer Research Society | CRS-24368 | John E Burke |
| Michael Smith Foundation for Health Research | 17686 | John E Burke |
| Canadian Institutes of Health Research | New Investigator | John E Burke |
| National Institutes of Health | GM132826 | Zied Gaieb<br>Rommie E Amaro |
| Stiftung FHNW | 341 | Matthias P Wymann |
| Swiss National Science Foundation | 310030_189065 | Matthias P Wymann |
| Novartis Foundation | 14B095 | Matthias P Wymann |
| Innosuisse - Schweizerische Agentur für Innovationsförderung | 37213.1 IP-LS | Matthias P Wymann |
| National Cancer Institute | P01-CA234228 | Zied Gaieb<br>Rommie E Amaro |

The funders had no role in study design, data collection and interpretation, or the decision to submit the work for publication.

### Author contributions

Manoj K Rathinaswamy, Conceptualization, Formal analysis, Investigation, Methodology, Writing - original draft, Writing - review and editing; Zied Gaieb, Software, Formal analysis, Validation, Investigation, Methodology, Writing - review and editing; Kaelin D Fleming, Data curation, Software, Formal analysis, Validation, Methodology, Writing - review and editing; Chiara Borsari, Resources, Investigation, Writing - review and editing; Noah J Harris, Investigation, Methodology, Writing - review and editing; Brandon E Moeller, Software, Methodology; Matthias P Wymann, Supervision, Funding acquisition, Validation, Writing - original draft, Writing - review and editing; Rommie E Amaro, Supervision, Funding acquisition, Methodology, Writing - review and editing; John E Burke, Conceptualization, Formal analysis, Supervision, Funding acquisition, Methodology, Writing - original draft, Project administration, Writing - review and editing

### Author ORCIDs

Manoj K Rathinaswamy (iD) https://orcid.org/0000-0001-5909-4353
Chiara Borsari (iD) http://orcid.org/0000-0002-4688-8362
Matthias P Wymann (iD) http://orcid.org/0000-0003-3349-4281
John E Burke (iD) https://orcid.org/0000-0001-7904-9859

### Decision letter and Author response

Decision letter https://doi.org/10.7554/eLife.64691.sa1
Author response https://doi.org/10.7554/eLife.64691.sa2

# Additional files

## Supplementary files

- Supplementary file 1. HDX-MS experimental conditions and data analysis parameters for *Figure 2* from the guidelines of the IC-HDX-MS community (*Masson et al., 2019*).

- Supplementary file 2. HDX-MS experimental conditions and data analysis parameters for *Figure 4* from the guidelines of the IC-HDX-MS community (*Masson et al., 2019*).

- Supplementary file 3. HDX-MS experimental conditions and data analysis parameters for *Figure 2—figure supplement 1* from the guidelines of the IC-HDX-MS community (*Masson et al., 2019*).

- Supplementary file 4. HDX-MS experimental conditions and data analysis parameters for *Figure 4—figure supplement 1* from the guidelines of the IC-HDX-MS community (*Masson et al., 2019*).

- Supplementary file 5. List of all PI3K inhibitors analysed in this manuscript. $IC_{50}$s for class IA and IB are listed from the reference attached. N.D. is not determined.

- Supplementary file 6. X-ray crystallography collection and refinement statistics.

- Transparent reporting form

## Data availability

The crystallography data has been deposited in the protein data bank with accession numbers (PDB: 7JWE, 7JX0, 7JWZ). The mass spectrometry proteomics data have been deposited to the ProteomeXchange Consortium via the PRIDE partner repository with the dataset identifier PXD021132. All data generated or analyzed during this study are included in the manuscript and supporting files. Specifically biochemical kinase assay data are included in the source data files.

The following datasets were generated:

| Author(s) | Year | Dataset title | Dataset URL | Database and Identifier |
|---|---|---|---|---|
| Rathinaswamy MK, Gaieb Z, Fleming KD, Borsari C, Harris NJ, Moeller BE, Wymann MP, Amaro RE, Burke JE | 2021 | Gedatolisib bound to the PI3Kg catalytic subunit p110 gamma | https://www.rcsb.org/structure/7JWE | RCSB Protein Data Bank, 7JWE |
| Rathinaswamy MK, Gaieb Z, Fleming KD, Borsari C, Harris NJ, Moeller BE, Wymann MP, Amaro RE, Burke JE | 2021 | NVS-PI3-4 bound to the PI3Kg catalytic subunit p110 gamma | https://www.rcsb.org/structure/7JX0 | RCSB Protein Data Bank, 7JX0 |
| Rathinaswamy MK, Gaieb Z, Fleming KD, Borsari C, Harris NJ, Moeller BE, Wymann MP, Amaro RE, Burke JE | 2021 | IPI-549 bound to the PI3Kg catalytic subunit p110 gamma | https://www.rcsb.org/structure/7JWZ | RCSB Protein Data Bank, 7JWZ |
| Rathinaswamy MK, Gaieb Z, Fleming KD, Borsari C, Harris NJ, Moeller BE, Wymann MP, Amaro RE, Burke JE | 2021 | Targeting of disease-linked mutations in PI3K gamma | https://www.ebi.ac.uk/pride/archive/projects/PXD021132 | PRIDE, PXD021132 |

The following previously published datasets were used:

| Author(s) | Year | Dataset title | Dataset URL | Database and Identifier |
|---|---|---|---|---|
| Tate JG, Bamford S, Jubb HC, | 2019 | COSMIC: the Catalogue Of Somatic Mutations In Cancer | https://cancer.sanger.ac.uk/cosmic | COSMIC, COSMIC |

| | | | | |
|---|---|---|---|---|
| Sondka Z, Beare DM, Bindal N, Boutselakis H, Cole CG, Creatore C, Dawson E, Fish P, Harsha B, Hathaway C, Jupe SC, Kok CY, Noble K, Ponting L, Ramshaw CC, Rye CE, Speedy HE, Stefancsik R, Thompson SL, Wang S, Ward S, Campbell PJ, Forbes SA | | | | |
| Murray JM, Ultsch M | 2017 | PI3K-gamma K802T in complex with Cpd 8 10-((1-(tert-butyl) piperidin-4-yl)sulfinyl)-2-(1-isopropyl-1H-1,2,4-triazol-5-yl)-5,6-dihydrobenzo[f]imidazo[1,2-d][1,4] oxazepine | https://www.rcsb.org/structure/6AUD | RCSB Protein Data Bank, 6AUD |
| Knight ZA, Gonzalez B, Feldman ME, Zunder ER, Goldenberg DD, Williams O, Loewith R, Stokoe D, Balla A, Toth B, Balla T, Weiss WA, Williams RL, Shokat KM | 2006 | A pharmacological map of the PI3-K family defines a role for p110alpha in signaling: The structure of complex of phosphoinositide 3-kinase gamma with inhibitor PIK-90 | https://www.rcsb.org/structure/2CHX | RCSB Protein Data Bank, 2CHX |
| Camps M, Ruckle T, Ji H, Ardissone V, Rintelen F, Shaw J, Ferrandi C, Chabert C, Gillieron C, Francon B, Martin T, Gretener D, Perrin D, Leroy D, Vitte P-A, Hirsch E, Wymann MP, Cirillo R, Schwarz MK, Rommel C | 2005 | Crystal structure of human PI3Kgamma complexed with AS604850 | https://www.rcsb.org/structure/2a4z | RCSB Protein Data Bank, 2A4Z |
| Elkins PA, Marrero EM | 2009 | Structure of Pi3K gamma with a potent inhibitor: GSK2126458 | https://www.rcsb.org/structure/3L08 | RCSB Protein Data Bank, 3L08 |

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
