## [Decision Letter]

**Acceptance summary:**

Your work is an excellent example of bringing a combination of powerful experimental and computational approaches together to gain significant insight into regulation of a class of enzymes (class 1 phosphoinositide 3-kinases) that are important drug targets in cancer and immunological disorders. This work examines the effects of both small molecule inhibitors and mutations that are known to activate and inactive the enzyme. The findings demonstrate similarities in the mechanisms by which drug compounds and genetic mutations shift the conformational ensemble to affect activity and will be useful as more effective therapies are sought to treat disease.

**Decision letter after peer review:**

Thank you for submitting your article "Disease related mutations in PI3Kγ disrupt regulatory C-terminal dynamics and reveals a path to selective inhibitors" for consideration by *eLife*. Your article has been reviewed by three peer reviewers, including Amy Andreotti as the Reviewing Editor and Reviewer #1, and the evaluation has been overseen by John Kuriyan as the Senior Editor.

The reviewers have discussed the reviews with one another and the Reviewing Editor has drafted this decision to help you prepare a revised submission.

Summary:

The structure of p110g was determined ~20 years ago but the ever-expanding knowledge of pi3kg biological role in disease keeps p110g as a worthy drug discovery target and focus of studies. The manuscript by Burke and colleagues describes a rigorous biochemical, structural and biophysical characterization of activating and inactivating mutations of pi3kg at R1021. Furthermore, the work highlights the mechanistic implications of the conformational changes exerted by known selective p110g inhibitors and diseased-linked mutation.

Overall this work presents a strong analysis of the structural basis of R1021 effects and makes important contributions to the field. There is, however, consensus among the reviewers that the paper needs to be significantly improved prior to publication.

Strengths of this work include:

HDX data that reveals that inhibitors that contact the activation loop are the ones that trigger allosteric changes.

The HDX signature of the R1021C mutant is similar to that of the enzyme bound to Gedatolisib, suggesting a shift to a similar conformational state in the mutant and inhibitor bound protein. Indeed, the increased potency of Gedatolisib for the mutant supports this observation. This aspect of the work highlights the importance of characterization of patient derived disease-causing mutations.

R1021 is a conserved residue in Class I PI3K so this work explores the interesting possibility of a conserved residue role in the design of selective inhibitors. Furthermore, mutations at this residue are displayed in tumors and in immunodeficient samples. The authors show how the R1021P from lowered expression yield to destabilization in molecular dynamic studies shows lowered kinase activity while the R1021C, display activation.

The study describes the changes observed in p110g upon binding to certain allosteric inhibitors and compared those to p110g/p101 complex to state that they are the same. It is interesting that inhibitors such as IPI-549 reduce exchange near the active site and protection in the hinge region. The inhibitor binds, as many of the selective inhibitor, with a propeller shape and pushing the orientation of M804. The group clustered with gedatolisib showed reduced exchange and increase exchange in kinase and helical domains. These changes are shown to be comparable to the ones off the partially activated of the R1021C mutant.

Inhibitor bound crystal structures have been solved that reveal features within the active site that are being exploited to achieve isoform specificity. To dissect the effect of the allosteric inhibitors the authors determined the structures with one of each class. The third cluster of inhibitors represented by NVS-PI34, shows a large conformational change that cannot occur in P110d or a and hence its isoform selectivity. The structures reveal the relative position of the inhibitors to the activation loop.

Another strength of this manuscript is the combination of synergistic structural analysis techniques that reveal both static as well as dynamic information especially relevant in the inhibitor binding analyses. The ability of X-ray crystallography to clearly define the drug binding interactions and HDX-MS to reveal dynamic allostery is quite powerful as together they allow for targeted inhibitor development and as the authors state, "…may reveal unique approaches to develop therapeutics for PI3K related human diseases.".

HDX-MS data revealed that the oncogenic, activating mutation R1021C showed localized increases in deuterium incorporation that suggest increases in backbone dynamics for the mutant PI3Kg when compared to the wild-type protein. Interestingly, the immunodeficiency, loss of function mutation R1021P despite the changes to deuterium incorporation being in the same locations as for R1021C, showed increases in deuterium incorporation that suggest not only increases in backbone dynamics in some cases, but also a disruption of the secondary structure. The authors also chose 7 inhibitors that were either selective for PI3Kg or pan-PI3K inhibitors for HDX-MS ligand binding analyses. Based on HDX-MS data the inhibitors were able to be divided into 3 groups that did not align with selectivity. They were able to be divided into those inhibitors where binding resulted in allosteric alterations to deuterium incorporation and those that only showed changes to deuterium incorporation within the inhibitor binding pocket. Intriguingly, the locations of the allosteric changes in HDX that were shown as a result of inhibitor binding were in comparable locations on the protein, albeit with variable magnitudes, as those seen with the R1021C mutation. X-ray crystallographic analysis of 3 of the inhibitors, described in the HDX-MS analyses, with PI3Kγ revealed new binding pockets.

Weakness: the paper is written in a way that makes it very difficult to fully appreciate (and integrate) these new observations. While this submission presents beautiful data, much of the importance is hidden or not well explained. A revised submission that clearly presents the connections between the range of approaches and results is needed to fully appreciate the efforts here to interrogate the mechanistic/allosteric features of this important enzyme.

In summary, the article presents many novel findings but it needs further analysis and corrections to complete the story.

Revisions:

Necessary improvements and clarifications:

The importance of the R1021 side chain and the tryptophan “lock” involving W1080 are nicely shown in Figure 1 however, in discussion of the HDX results (and MD simulations) the effect of mutation at R1021 on the region surrounding the tryptophan lock is not clearly presented. It is not until the Discussion section that the connection is briefly revisited; the data could be emphasized earlier to make this important point more clearly early on.

In another example, the activity data provided in Figure 1 are quite interesting and yet the authors do not adequately come back to this data once the HDX data has been analyzed to provide further insight into some of the surprising effects on activity especially with respect to ATPase versus lipid phosphorylation. Since at the end of the day it is the changes in enzyme activity that likely lead to disease phenotype the connections between the biophysical data and the functional assays should be strengthened (especially for the R1021C mutation that does not appear to suffer from poor expression).

Connections between WT binding to membrane and intrinsic behavior of the R1021C mutant (measured by HDX) seem really interesting but are not clearly presented. Overall, the authors seem to be concluding that the R1021C mutation might preorganize the enzyme for catalysis. A clearer presentation integrating the data (functional assays, HDX, and MD simulation results) that support this idea would greatly strengthen this paper.

Overall, the manuscript reads as a series of unrelated experiments with connections between the observations from different techniques lacking or at least not clearly stated. The authors should be able to more clearly connect the characterization of the mutants and inhibitor binding – the data is excellent and therefore deserves an improved narrative.

One general comment that might help emphasize the important findings is that at present the paper seems to give equal time to both the C and P mutations at position 1021. Even in the paragraph that's focused on R1021C mechanistic insights, mention of the destabilizing effect of the proline mutation is included. The authors may have reason to stay with this strategy but it is clear that the proline mutation disrupts the structure and so in the mind of this reviewer the R1021C mutation is much more interesting from an allosteric point of view. Focusing on the details of the allosteric mechanism for the R1021C mutation might help bring out the importance of this work.

The experiments using HDX-MS carried out in p110g/p101 and its variants are elegant. The authors should include a biological or just experimental rational of whether they attempted complexes with p84 or why they skipped them.

A figure of a sequence alignment with p110a and p110d in the area of p110g-R1021 will clarify this residue in comparison with mutations studied in the other isoforms. This should be early on in the manuscript (Introduction, first section of Results). the fact that the equivalent residues in p110a (992) is found in cancer should be included too.

The results state that the mol dyn studies of the R1021C and R1021P mutants increase instability of p110g (text and Figure 3). The author should elaborate further whether they refer to increase thermal mobility or mobility, and or instability. It is hard to follow a rational where instability confers increase activity. It is noted that, as written, the molecular dynamics simulations do not seem to add significant information that is not already able to be gleaned from the HDX-MS and X-ray data.

Fix sentence in Introduction paragraph one, the catalytic subunits are p110a etc never PI3Ka, sentence is not clear

Introduction paragraph two, which act as negative regulator or keep pi3k autoinhibited would be better language than. That the regulatory domain are potent inhibitors.

Introduction paragraph four, the activating mutations clusters to the c-terminal regulatory motif of pi3k should be p110 (iis off the catalytic subunit, right?) and in the event that they want to refer to both p110 and p84 it should be explicit

Figure 2 AB should be labeled p110g and in the caption mentioned that the results are from hdx done with the Complex. The title p110g/p101 on the ribbons that show only p110g are misleading

Figure 6 has the potential to be a key figure but needs work to make it clear.

the title inactive / active are not clear should they be to the left

might be that the closed c-terminal regulatory is at the center of the figure (inactive) and then ways to activated irradiate with 3 arrows one to the p mutation, another to c mut, another to the Inhibitor

The section conformational selective inhibitors show altered specificity toward PI3kg mutant needs to be developed. The authors showed that R1021c mutant display similar conformational changes to wt-gedatolisib and they conclude that there would be altered inhibitor binding. This is not clear and the conclusion should be further rationalized.

Discussion paragraph two should state in the regulatory motif described in the kinase domain to be clear

In an attempt to strive for reproducibility the authors should include the size of the crystals, the size of the beam used for data collection as well as the method of data collection (single shot, vector, etc).

The authors worked on 3 structures at different ranges of resolution. It would help the reader to add the pdb id in the table below the title of the complex as well as a +. Although the complex with NVS-PI3-4 is ok in the absence of waters (resolution 3.15); it is weird that the structure with IPI549 at 2.65 does not have waters. Is this a typo? If it is not, an explanation should be included. Moreover, the reported rmsd of bond length and bond angles are too small below a typical deviation of at least 0.008 for rmsd of bond length. The structure should be re-refined lowering the weights of the geometrical restrains. The comment that the first structure was refined from one crystal should be elaborated. The other 2 where collected from multiple crystals to get one data set, or multiple data sets were collected and refined and only one is shown here? The authors should provide the validation report from the deposition in the PDB.

General HDX-MS comments:

The supplemental excel data file with the HDX data needs more information. Is it possible to add as additional columns in each sheet:

-the raw deuterium incorporation data

-the maximum incorporation per peptide

-peptide coverage maps

-indications on the list of peptides those peptides that were chosen to display on the cartoon structures in each figure

The HDX supplemental table S1 is very difficult to read as there is so much information in one page. I can empathize with the complexity of this table considering HDX-MS studies such as this, so would it be better to break out this one huge table into smaller tables that would go with each experiment. These individual tables could also go within the supplemental excel file within the tab that hold the corresponding HDX-MS data.

These HDX comments stem from Figure 2 and also apply to the other HDX figures as well:

How are the data "corrected" for back exchange? Can this be explained or described explicitly in the Materials and methods section? Is it possible to list the equation that was used? I am concerned with flat correction for back exchange based on the percent deuterium in the labeling reaction for a number of reasons, not the least of which is that not all peptides have the same level of back exchange. Do you have an idea of the back exchange in your system in general? Without more information about how your data were corrected, I view the presentation of the data, as is, as problematic. If there was no true maximal deuteration control collected, and without justification as to how the data were analyzed, I would prefer to see the uncorrected data plotted in panels A and B. I understand that there is a difference in percent deuterium in the labeling reactions across the experiments, but I need more information at this point.

It isn't clear which time point was used for difference data plotted in panels A and B. Can you be more specific in the method how the peptides were chosen and what exactly is plotted in the figure panels.

How are the data in panels C, D, E, and F calculated? Can this be explained or described explicitly in the Materials and methods section? Can the data in these panels also be included in the supplemental excel file? Additionally, these panels are also VERY small and were most useful when at a high zoom on the screen. Is it possible to increase the sizes to make them more accessible? I wonder also if it would be better to not have these data as the aggregate of all time points and instead show all the timepoints individually in each panel? This would be helpful in the discussion of the different types of change to the deuterium incorporation, be it a change in dynamics or secondary structure, etc.

Other HDX figure specific comments:

Figure 2—figure supplement 1. The increase in HDX changes shown here in this figure, I assume, are due to the loss of the stabilization that is provided by binding of p101. Is this increase in HDX difference also seen for the wild-type protein? Specifically, is the difference you would calculate for: p110g R1021C- p110g R1021C:p101 the same as for: p110g -p110g:p101?

Figure 4—figure supplement 1. There is something wonky about the legend on this figure. There are descriptions for panels a through D which don't match the figures panels A through G.

---

## [Author Response]

Revisions:Necessary improvements and clarifications:The importance of the R1021 side chain and the tryptophan “lock” involving W1080 are nicely shown in Figure 1 however, in discussion of the HDX results (and MD simulations) the effect of mutation at R1021 on the region surrounding the tryptophan lock is not clearly presented. It is not until the Discussion section that the connection is briefly revisited; the data could be emphasized earlier to make this important point more clearly early on.

We agree that it is important to highlight the importance of the interactions between R1021 with the regulatory motif, and how this alters the stability of the tryptophan lock. We have clarified the connection between the molecular interactions of the R1021 residue and the MD and HDX results. We have integrated this at multiple locations in the Results section. This is emphasized by changes made in the text.

In another example, the activity data provided in Figure 1 are quite interesting and yet the authors do not adequately come back to this data once the HDX data has been analyzed to provide further insight into some of the surprising effects on activity especially with respect to ATPase versus lipid phosphorylation. Since at the end of the day it is the changes in enzyme activity that likely lead to disease phenotype the connections between the biophysical data and the functional assays should be strengthened (especially for the R1021C mutation that does not appear to suffer from poor expression).

We have added additional discussion of the activity data for the different p110 constructs and how they affect biochemical activity, both basal ATPase, and lipid phosphorylation. We have linked the difference in biochemical activity to the observations in the biophysical experiments.

Connections between WT binding to membrane and intrinsic behavior of the R1021C mutant (measured by HDX) seem really interesting but are not clearly presented. Overall, the authors seem to be concluding that the R1021C mutation might preorganize the enzyme for catalysis. A clearer presentation integrating the data (functional assays, HDX, and MD simulation results) that support this idea would greatly strengthen this paper.

We agree with the reviewers assessment that the R1021C mutant appears to be pre-organising the enzyme for membrane binding. We have further elaborated on this in both the Results and Discussion sections.

Overall, the manuscript reads as a series of unrelated experiments with connections between the observations from different techniques lacking or at least not clearly stated. The authors should be able to more clearly connect the characterization of the mutants and inhibitor binding – the data is excellent and therefore deserves an improved narrative.One general comment that might help emphasize the important findings is that at present the paper seems to give equal time to both the C and P mutations at position 1021. Even in the paragraph that's focused on R1021C mechanistic insights, mention of the destabilizing effect of the proline mutation is included. The authors may have reason to stay with this strategy but it is clear that the proline mutation disrupts the structure and so in the mind of this reviewer the R1021C mutation is much more interesting from an allosteric point of view. Focusing on the details of the allosteric mechanism for the R1021C mutation might help bring out the importance of this work.

We appreciate the reviewers careful assessment of the manuscript, and have made multiple additions to connect the experiments spanning kinase assays, HDX-MS, MD, and X-ray approaches. We have linked the results from these different experiments at the beginning and ending of each section, so that the paper more clearly flows through each experiment. As suggested, we have also minimised our discussion of the R1021P, as we agree that the R1021C mutant is the most conceptually interesting alteration.

The experiments using HDX-MS carried out in p110g/p101 and its variants are elegant. The authors should include a biological or just experimental rational of whether they attempted complexes with p84 or why they skipped them.

The manuscript is focused on the regulatory aspects of the p110 subunit. The rationale behind using p110/p101 was to show the mutation/inhibitor effects occurring in a biologically relevant context of a heterodimeric complex. We have previously mapped the p110γ/p84 (Walser et al., 2013) and p110γ/p101 (Vadas et al., 2013) interfaces using HDX-MS, which showed no changes in the regulatory motif.

As both p84 or p101 complexes with p110 do not affect the region under study, studying only the p101 complex is sufficient to establish biological relevance. Hence, we believe that studies on the effects of these mutations in the context of both complexes would be redundant.

A figure of a sequence alignment with p110a and p110d in the area of p110g-R1021 will clarify this residue in comparison with mutations studied in the other isoforms. This should be early on in the manuscript (Introduction, first section of Results). the fact that the equivalent residues in p110a (992) is found in cancer should be included too.

We agree that an alignment will be informative as to the conservation of this region in other p110 isoforms. The sequence alignment of ka8 of all class I p110 isoforms has now been added to Figure 1 as panel C. We have also discussed the alignment and oncogenic mutant in PIK3CA in the first paragraph of the Results.

The results state that the mol dyn studies of the R1021C and R1021P mutants increase instability of p110g (text and Figure 3). The author should elaborate further whether they refer to increase thermal mobility or mobility, and or instability. It is hard to follow a rational where instability confers increase activity. It is noted that, as written, the molecular dynamics simulations do not seem to add significant information that is not already able to be gleaned from the HDX-MS and X-ray data.

The MD studies revealed disruption in the local hydrogen bonding network in the regulatory motif, and altered orientation of the helices in the regulatory motif compared to WT. In the Results section we specifically focus only on these two factors, and do not claim global changes to protein stability. This is likely due to the timescale of the simulations (~2-3 µs). This point has been clarified in the text. However, this data still supports a role of the R1021C mutant in locally destabilising the regulatory motif, consistent with the HDX-MS and kinase data. We have added in additional details in the text.

Fix sentence in Introduction paragraph one, the catalytic subunits are p110a etc never PI3Ka, sentence is not clear

The sentence – and the rest of the manuscript – has been corrected and now only refers to the isolated catalytic subunits as p110s. In situations where we do not know the composition of the complex (ie for biological experiments where inhibition can occur for either p110γ/p101 or p110γ/p84 we refer to this as PI3Kγ, consistent with conventions in the PI3K field).

Introduction paragraph two, which act as negative regulator or keep pi3k autoinhibited would be better language than. That the regulatory domain are potent inhibitors.

We have removed this sentence as this point might be distracting to the rest of the paragraph, as linking to class IA here is confusing.

Introduction paragraph four, the activating mutations clusters to the c-terminal regulatory motif of pi3k should be p110 (iis off the catalytic subunit, right?) and in the event that they want to refer to both p110 and p84 it should be explicit

We have changed this wording to more clearly focus on the p110 catalytic subunit.

Figure 2 AB should be labeled p110g and in the caption mentioned that the results are from hdx done with the Complex. The title p110g/p101 on the ribbons that show only p110g are misleading

We have updated the figure/legend as suggested.

Figure 6 has the potential to be a key figure but needs work to make it clear.the title inactive / active are not clear should they be to the leftmight be that the closed c-terminal regulatory is at the center of the figure (inactive) and then ways to activated irradiate with 3 arrows one to the p mutation, another to c mut, another to the Inhibitor

The figure has now been modified to reflect the recommended changes from the reviewer.

The section conformational selective inhibitors show altered specificity toward PI3kg mutant needs to be developed. The authors showed that R1021c mutant display similar conformational changes to wt-gedatolisib and they conclude that there would be altered inhibitor binding. This is not clear and the conclusion should be further rationalized.We have added additional description of the rationale for this experiment, and described potential mechanisms underlying the sensitivity of the R1021C mutant to the Gedatolisib.Discussion paragraph two should state in the regulatory motif described in the kinase domain to be clearWe have fixed this for clarityIn an attempt to strive for reproducibility the authors should include the size of the crystals, the size of the beam used for data collection as well as the method of data collection (single shot, vector, etc).We have included all of the requested methodological details for crystals in the Materials and methods section, and our data collection strategy.The authors worked on 3 structures at different ranges of resolution. It would help the reader to add the pdb id in the table below the title of the complex as well as a +. Although the complex with NVS-PI3-4 is ok in the absence of waters (resolution 3.15); it is weird that the structure with IPI549 at 2.65 does not have waters. Is this a typo? If it is not, an explanation should be included. Moreover, the reported rmsd of bond length and bond angles are too small below a typical deviation of at least 0.008 for rmsd of bond length. The structure should be re-refined lowering the weights of the geometrical restrains. The comment that the first structure was refined from one crystal should be elaborated. The other 2 where collected from multiple crystals to get one data set, or multiple data sets were collected and refined and only one is shown here? The authors should provide the validation report from the deposition in the PDB.

We apologise for a lack of details in the original manuscript that may have led to some confusion in the analysis of the X-ray crystallography data. We have added the PDB identifiers to the table as requested, added the PDB validation reports, and expanded our discussion of the crystallographic methods.

As for the concern about the lack of waters in the structures, this is most likely a concern based on how we selected our resolution cut-offs. We attempted to include as much high resolution data as possible based on the CC1/2 cutoff (~0.4) based on the protocol of Karplus et al. Science 2012. The I/Io in our high resolution shell is relatively low (ranging from 0.7-0.9) so our resolution estimates may be 0.1-0.2 lower than what is classically reported. This is right at the resolution where waters may be visible. We re-refined the IPI-549 structure where we attempted both automated (in Phenix) and manual addition of water (in Coot). The addition of waters led to an increased Rfree values, and for this reason we feel that modeling water in this structure is not appropriate. We have added description of our resolution cut-offs in the Materials and methods.

**Author response image 1. sa2fig1:** 

Finally we utilised the standard geometrical restraints within Phenix at the final step of refinement. As suggested, we re-refined the structure with variations in the geometrical restraints, however, this always led to increased Rfree values, and worse/similar model parameters (Ramachandran, rotamers, Clashscore, and r-work, see picture of an example refinement for the IPI-549 complex and note the statistics compared to the original statistics). Low RMS values for bonds and angles is noted as an output feature of Phenix at medium resolution. The following text is taken directly from the Phenix documentation(see https://www.phenix-online.org/documentation/faqs/refine.html):

“The RMS(bonds) and RMS(angles) in the output model are too low after weight optimization – how can I make them higher?

This is a common misconception about geometry deviations, based partly on anecdotal experience with other refinement programs. The target RMSDs come from looking at very accurate high-resolution small molecule structures, so they reflect the real variation that should occur in geometry. At lower resolution, even though you know that the bond lengths and bond angles must vary as much as they do in high-resolution structures, there isn’t enough experimental data to tell in which direction they should deviate from the expected values. So it is reasonable for the refined RMSDs to be lower than the targets. (Anecdotally, we have found that phenix.refine often refines to much tighter RMSDs with similar R-frees to other programs. This may reflect different approaches used in the geometry restraints, X-ray target, or optimization methods, but it should not be cause for concern.)”

For this reason, we consider the current models appropriate, and do not see an advantage of altering the refinement parameters for the final model.

Finally, all crystal structures were solved from diffraction data from a single crystal. These details, as well as more complete data collection statistics, have been added to the Materials and methods, with validation reports from the PDB included in the submission documents.

General HDX-MS comments:The supplemental excel data file with the HDX data needs more information. Is it possible to add as additional columns in each sheet:-the raw deuterium incorporation data-the maximum incorporation per peptide-peptide coverage maps-indications on the list of peptides those peptides that were chosen to display on the cartoon structures in each figure

We have added all of the requested information to the source data file. In addition we have added the raw HDExaminer output for all experiments. We have also added Peptide coverage maps to the source data file for each HDX figure. Finally we have added in the %D difference values in the source data, and highlighted all peptides with significant differences that were mapped on the cartoon structures in each figure.

The HDX supplemental table S1 is very difficult to read as there is so much information in one page. I can empathize with the complexity of this table considering HDX-MS studies such as this, so would it be better to break out this one huge table into smaller tables that would go with each experiment. These individual tables could also go within the supplemental excel file within the tab that hold the corresponding HDX-MS data.

The original table S1 has been split into multiple tables (now Supplementary files 1-4), with each table corresponding to each main/supplemental figure.

These HDX comments stem from Figure 2 and also apply to the other HDX figures as well:How are the data "corrected" for back exchange? Can this be explained or described explicitly in the Materials and methods section? Is it possible to list the equation that was used? I am concerned with flat correction for back exchange based on the percent deuterium in the labeling reaction for a number of reasons, not the least of which is that not all peptides have the same level of back exchange. Do you have an idea of the back exchange in your system in general? Without more information about how your data were corrected, I view the presentation of the data, as is, as problematic. If there was no true maximal deuteration control collected, and without justification as to how the data were analyzed, I would prefer to see the uncorrected data plotted in panels A and B. I understand that there is a difference in percent deuterium in the labeling reactions across the experiments, but I need more information at this point.

We apologise for this confusion in the original version. There is no correction for back exchange in this dataset, as the generation of a fully deuterated protein for class I PI3Ks has been very challenging. We meant to state that the only correction is for the amount of deuterium in the buffer, and that there is no correction for back exchange. The data as submitted in the source data is the raw uncorrected data. Experiments on previous fully deuterated control protein has revealed back exchange levels ranging from 10-45%, with a mean value of ~25%.

This has now been addressed in the Materials and methods section.

It isn't clear which time point was used for difference data plotted in panels A and B. Can you be more specific in the method how the peptides were chosen and what exactly is plotted in the figure panels.

For panels A+B we mapped changes on the structure for any peptide that crossed our significance threshold at any time point. This is consistent with the most simplified method to represent HDX data. To clarify this point, we have highlighted in the source data file the %D difference at every time point.

How are the data in panels C, D, E, and F calculated? Can this be explained or described explicitly in the Materials and methods section? Can the data in these panels also be included in the supplemental excel file? Additionally, these panels are also VERY small and were most useful when at a high zoom on the screen. Is it possible to increase the sizes to make them more accessible? I wonder also if it would be better to not have these data as the aggregate of all time points and instead show all the timepoints individually in each panel? This would be helpful in the discussion of the different types of change to the deuterium incorporation, be it a change in dynamics or secondary structure, etc.

The data in these panels are generated by taking the sum of the difference between conditions at all time points. It is now clearly described in the Materials and methods section. There are some advantages in showing a graph for each time point as described by the reviewer, however, for very data intensive figures (ie the inhibitor data in Figure 4A), this is very confusing, as these panels would have 12 curves. The simplest representation for these data is to have #D difference as a sum, and then for important regions/peptides show the individual peptide graphs (ie Figure 2G+4E), as this has the individual timepoint differences.

This has been clarified in the Materials and methods section and the graph size has been increased.

Other HDX figure specific comments:Figure 2—figure supplement 1. The increase in HDX changes shown here in this figure, I assume, are due to the loss of the stabilization that is provided by binding of p101. Is this increase in HDX difference also seen for the wild-type protein? Specifically, is the difference you would calculate for: p110g R1021C- p110g R1021C:p101 the same as for: p110g -p110g:p101?

The reviewer is correct that the larger magnitude in HDX changes, that occurs for the free p110γ subunit with the mutation are due to the free p110γ subunit being greatly more dynamic in absence of the p101 subunit. The differences between p110γ and p110γ-p101 was previously published by us in Vadas et al., 2013.

As for the reviewers question on the stabilising role of the p101 subunit between the WT and mutant R1021C, we have included new HDX-MS peptide graphs comparing all 4 conditions. This revealed that the differences are greater for the R1021C mutant with p101 when compared to the WT protein. See new Figure 2—figure supplement 1 (previously Figure S3), and description in the Results section.

Figure 4—figure supplement 1. There is something wonky about the legend on this figure. There are descriptions for panels a through D which don't match the figures panels A through G.

We apologise for this oversight, and have now corrected Figure 4—figure supplement 1 and the figure legend.